# Neutrophil extracellular trap formation requires OPA1-dependent glycolytic ATP production

Poorya Amini[1], Darko Stojkov[1], Andrea Felser [2], Christopher B. Jackson [3], Carolina Courage[4], André Schaller [4], Laurent Gelman[5], Maria Eugenia Soriano [6], Jean-Marc Nuoffer [2], Luca Scorrano [7], Charaf Benarafa [8,9], Shida Yousefi [1] & Hans-Uwe Simon [1]

Optic atrophy 1 (OPA1) is a mitochondrial inner membrane protein that has an important role in mitochondrial fusion and structural integrity. Dysfunctional OPA1 mutations cause atrophy of the optic nerve leading to blindness. Here, we show that OPA1 has an important role in the innate immune system. Using conditional knockout mice lacking Opa1 in neutrophils (*Opa1*[NΔ]), we report that lack of OPA1 reduces the activity of mitochondrial electron transport complex I in neutrophils. This then causes a decline in adenosine-triphosphate (ATP) production through glycolysis due to lowered $NAD^+$ availability. Additionally, we show that OPA1-dependent ATP production in these cells is required for microtubule network assembly and for the formation of neutrophil extracellular traps. Finally, we show that *Opa1*[NΔ] mice exhibit a reduced antibacterial defense capability against *Pseudomonas aeruginosa*.

[1] Institute of Pharmacology, University of Bern, 3010 Bern, Switzerland. [2] University Institute of Clinical Chemistry, Bern University Hospital, 3010 Bern, Switzerland. [3] Research Program for Molecular Neurology, Biomedicum Helsinki, University of Helsinki, 00290 Helsinki, Finland. [4] Division of Human Genetics and Department of Pediatrics, Inselspital, Bern University Hospital, University of Bern, 3010 Bern, Switzerland. [5] Friedrich Miescher Institute for Biomedical Research, 4058 Basel, Switzerland. [6] Department of Biology, University of Padua, 35121 Padua, Italy. [7] Venetian Institute of Molecular Medicine (VIMM), 35129 Padua, Italy. [8] Institute of Virology and Immunology, 3147 Mittelhäusern, Switzerland. [9] Department of Infectious Diseases and Pathology, Vetsuisse Faculty, University of Bern, 3012 Bern, Switzerland. These authors contributed equally: Poorya Amini, Darko Stojkov. Correspondence and requests for materials should be addressed to H.-U.S. (email: hus@pki.unibe.ch)

Besides intracellular killing, neutrophils are able to exert an antibacterial activity in the extracellular space by the formation of so-called neutrophil extracellular traps (NETs)[1,2]. NETs consist of DNA and granule proteins and are able to bind and to kill extracellular pathogens. Thus, NETs represent an important element of the innate immune response and they are seen in association with many infectious, allergic, and autoimmune diseases[3–5]. It should be noted that such extracellular DNA traps able to kill bacteria can also be formed by eosinophils[6] and basophils[7]. Although the functional importance of NETs is generally accepted, the origin of the DNA scaffold in NETs, as well as the mechanism of their generation, remains unclear and a matter of dispute.

Since we had observed that neutrophils and other granulocytes release mitochondrial DNA (mtDNA) in the formation of extracellular DNA traps[3], we hypothesized that mitochondrial dynamics might be important for mtDNA release. Five members of the GTPase dynamin family, namely mitofusin-1 (MFN1), mitofusin-2 (MFN2), optic atrophy 1 (OPA1), mitochondrial fission 1 (FIS1), and dynamin-related protein 1 (DRP1) are known as the "mitochondria-shaping" proteins which regulate the fusion and fission of mitochondria[8]. OPA1 is anchored to the mitochondrial inner membrane, and, together with MFN1 and MFN2, which are located in the outer mitochondrial membrane, it controls the mitochondrial fusion process[9]. Interestingly, and independent of its role in mitochondrial fusion, OPA1 is also responsible for maintaining the mitochondrial cristae junctions tight, thus allowing efficient oxidative respiration[10–12]. Genetic depletion of OPA1 resulted in reduced cell growth and lowered mitochondrial membrane potential, as well as in defective cellular respiration and metabolism[13,14]. Patients with mutated OPA1 can develop autosomal dominant optic atrophy (ADOA), which leads to loss of bilateral visual function within the first two decades of life, or may present with additional signs of peripheral neuropathy, deafness, cerebellar ataxia, paraparesis, and myopathy[15,16]. Reduction of OPA1 activity has also been associated with other neurodegenerative diseases, such as Alzheimer, Huntington, and Parkinson diseases[17].

In this paper, we demonstrate that the lack of OPA1, or mutations in the OPA1 gene, can also result in functional defects outside of the nervous system. Specifically, we show that OPA1 is required for ATP production through glycolysis in neutrophils, cells which are known to contain a modest number of mitochondria, but are unable to perform mitochondrial respiration[18]. ATP production in neutrophils is required for the assembly of the microtubule network. Furthermore, this network has recently been demonstrated to be essential for NET production[19]. Neutrophils lacking other members of GTPase dynamin family exhibit no such defect in NET formation, suggesting that the OPA1 effects on neutrophils are a consequence of its role in maintaining the mitochondrial cristae junctions tight and thus likely to be independent of mitochondrial fusion. Moreover, mice with Opa1 knockout neutrophils demonstrate defects in antibacterial defense owing to their inability to form NETs under in vivo conditions. Taken together, the findings presented here provide a significant step forward in understanding OPA1 mitochondrial functions in non-neuronal cells and extend the impact of these functions to the innate immune system.

## Results

**OPA1-deficient neutrophils are unable to form NETs.** To test whether members of GTPase dynamin family are involved in the process of extracellular DNA release from neutrophils, we used a small hairpin RNA (shRNA)-based approach. To this end, lentivirus particles containing five different human shRNA sequences for each candidate gene, namely; DRP1, FIS1, MFN1, MFN2, and OPA1 (Mission shRNA, Sigma-Aldrich) were transduced into human myeloid leukemia cells (PLB-985). Similarly, mouse shRNAs were used in Hoxb8-immortalized mouse progenitor cells (Hoxb8 neutrophils)[20]. After antibiotic selection and cloning of single cells, transduced clones were tested for the down-regulation of target genes using qPCR (Supplementary Fig. 1a, b, left panels). Cells with the lowest expression of target genes were differentiated to mature neutrophils and examined for their ability to release DNA following GM-CSF priming and subsequent C5a stimulation[21]. Reduction of OPA1 mRNA expression, but not any of the other dynamin-like GTPases, was associated with a complete inability to release DNA with either differentiated human myeloid leukemia cells or mouse Hoxb8 neutrophils (Supplementary Fig. 1a, b, right panels).

To verify the relevance of OPA1 for DNA release and NET formation in primary human neutrophils, we isolated blood neutrophils from two patients of the same family (father and son), both suffering from autosomal dominant optic atrophy disease (ADOA). They harbor the heterozygous synonymous mutation c.1140G>A/ p.(Glu380Glu) resulting in exon 11 skipping in the original isoform identified (NM_015560.2) (Fig. 1a)[22]. While the neutrophils of the father released extracellular DNA following combined GM-CSF/C5a stimulation in the same way as control neutrophils, activation of neutrophils from the son failed to cause DNA release (Fig. 1b), in spite of their similar level of OPA1 protein expression (Fig. 1c). Although it remains unclear why the neutrophils of father and son behaved differently, we concluded that mutated OPA1 can result in a functional defect with respect to extracellular DNA release in primary human neutrophils.

It has been speculated that an OPA1 deficiency might cause mtDNA instability in neurons, leading to reduced mtDNA levels[23,24]. To test this hypothesis, we compared the mtDNA content by analyzing the levels of the mitochondrial gene ATP synthase protein 8 (MT-ATP8) and the mitochondrial D-loop region compared to the single copy β2 microglobulin (B2M) nuclear gene[25] and observed no differences between neutrophils of ADOA patients and healthy controls (Fig. 1d). These findings are in agreement with a previous report demonstrating that white blood cells of ADOA patients exhibit a normal content of mtDNA[26]. Therefore, it seems unlikely that the failure to release DNA in OPA1 defective neutrophils is the consequence of reduced mtDNA levels.

Since ADOA exhibits heterogeneous phenotypes[12,27] and as it is a rare disease, making it difficult to obtain primary neutrophils from patients, we pursued a genetic approach to investigate more closely the role of OPA1 in DNA release and NET formation. Mouse and fly models that carry homozygous Opa1 mutations are embryonically lethal[28]. Therefore, we generated $Opa1^{flox/flox}$-$Lyz2^{Cre/Cre}$ mice (designated $Opa1^{N\Delta}$ mice hereafter) to study the effects of Opa1 deficiency in neutrophils (Supplementary Fig. 2a). Mature bone marrow neutrophils of $Opa1^{N\Delta}$ mice displayed a complete OPA1 knockout at the protein level (Supplementary Fig. 2b). In contrast to neutrophils from control $Lyz2^{Cre/Cre}$ mice, $Opa1^{N\Delta}$ primary mouse neutrophils did not exhibit any detectable DNA release upon GM-CSF priming and activation with C5a, with LPS treatment, or following co-culture with E. coli–GFP (Fig. 1e, left panel). To quantify the dsDNA released by activated neutrophils, we collected the culture supernatants and measured the amount released in the supernatant using Pico-Green fluorescent dye. In contrast to $Lyz2^{Cre/Cre}$ mice, neutrophils derived from $Opa1^{N\Delta}$ mice were unable to release dsDNA upon activation (Fig. 1e, right panel). We also tested other activating reagents previously reported to induce NET formation such as PMA, P. aeruginosa in co-culture, or nucleic

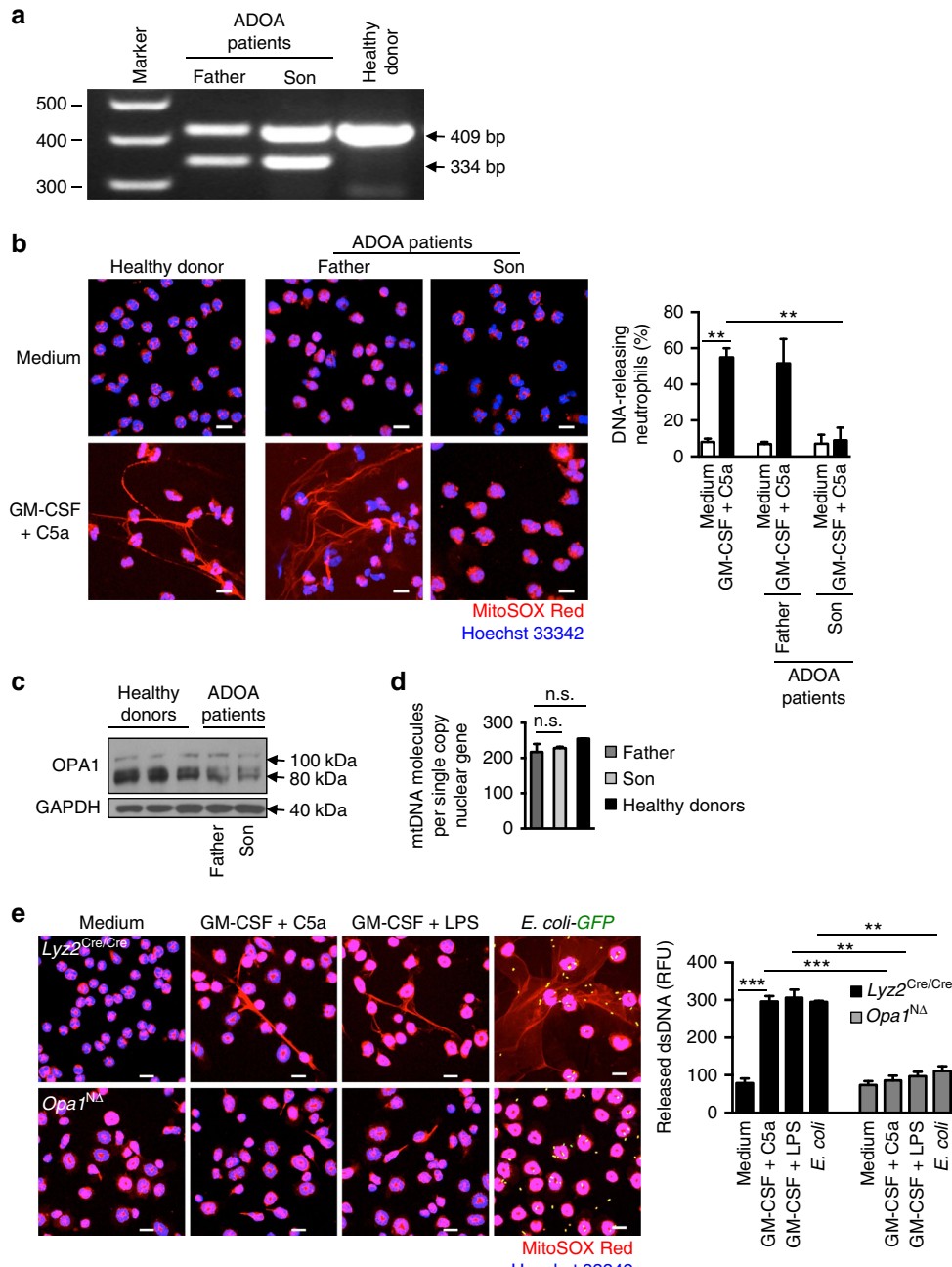

**Fig. 1** Failure of DNA release by OPA1-deficient human and mouse neutrophils. **a** Molecular characterization of *OPA1* transcripts in ADOA patients. OPA1 transcript analysis from isolated neutrophils encompassing exons 10–12 derived from a control and two ADOA patients harboring the heterozygous c.1140G>A mutation (NM_015560.2). A shortened product of 334 bp reveals the aberrantly spliced, skipped exon 11 and is detected in patients' cDNA only. The band of 409 bp indicates the wild-type transcript. A full image of the agarose gel is provided in Supplementary Fig. 11a. **b** Confocal microscopy. Highly purified human blood neutrophils from control individuals and two ADOA patients were primed with GM-CSF and subsequently stimulated with C5a. Extracellular DNA was stained with MitoSOX™ Red and the nucleus with Hoechst 33342 (blue). Bars, 10 μm. Right: Quantification of the number of DNA-releasing neutrophils. Values are means ± SEM. **$p < 0.01$; $n = 3$. **c** Immunoblotting. Protein lysates of freshly isolated neutrophils from ADOA patients and healthy donors were analyzed for OPA1 protein expression. Full-length immunoblots are provided in Supplementary Fig. 11b. **d** Quantitative PCR. Genomic DNA from freshly purified human neutrophils of ADOA patients and healthy control donors were analyzed for mtDNA content (average of ATP synthase protein 8 (*MT-ATP8*) and mitochondrial D-Loop) per single copy nuclear gene (*B2M*). Values are means ± SEM ($n = 3$). **e** Confocal microscopy. Primary mature neutrophils from *Opa1*NΔ and control mice were primed with GM-CSF and subsequently stimulated with C5a, LPS, or co-cultured with *E. coli*–GFP. Extracellular DNA was stained with MitoSOX™ Red and the nucleus with Hoechst 33342 (blue). Bars, 10 μm. Right: quantification of released dsDNA in supernatants of activated neutrophils. Values are means ± SEM. n.s., not significant; **$p < 0.01$; ***$p < 0.001$; $n = 5$. Additional data including NET formation induced by other triggers are provided in Supplementary Fig. 2c

acid–containing immune complexes (ICs) produced by combining the small nuclear ribonucleoprotein (smRNP) antigen with either SLE sera (RNP ICs-SLE) or with anti-damaged-DNA/RNA antibody (RNP-ICs-Ab)[29]. None of these stimuli could trigger NET formation in $Opa1^{N\Delta}$ neutrophils (Supplementary Fig. 2c). To exclude any effect of Cre recombinase on extracellular DNA release, we also analyzed $Opa1^{flox/flox}$ neutrophils and observed no difference as compared to $Lyz2^{Cre/Cre}$ neutrophils (Supplementary Fig. 2d).

We next investigated functional consequences of the lack of DNA release in $Opa1^{N\Delta}$ neutrophils in terms of NET formation. Since granule proteins are essential components of NETs[2,21,30], we performed immunofluorescence microscopy studies in which we searched for the presence of elastase within the released DNA scaffold of activated mouse neutrophils. In activated control $Lyz2^{Cre/Cre}$ neutrophils, we found extracellular DNA fibers colocalizing with elastase (Supplementary Fig. 2e). In contrast, with $Opa1^{N\Delta}$ neutrophils, we were unable to detect elastase-containing extracellular structures.

In summary, we report that neutrophilic cell lines and primary human or mouse neutrophils deficient in functional OPA1 are unable to form the extracellular DNA scaffolds as seen in NETs.

**OPA1-deficient neutrophils exhibit disruption of the microtubule network.** Mitochondrial localization analysis revealed a wide distribution in the cytoplasm of resting $Opa1^{N\Delta}$ neutrophils, whereas in control $Lyz2^{Cre/Cre}$ neutrophils, mitochondria were concentrated in the perinuclear region (Supplementary Fig. 3a). Following combined GM-CSF/C5a stimulation of control neutrophils, the vast majority of mitochondria translocated toward the plasma membrane. In contrast, in stimulated $Opa1^{N\Delta}$ neutrophils, mitochondria remained widely distributed showing no change in location (Supplementary Fig. 3a). This finding points to a possible defect in microtubule network dynamics[31]. Further suggestive of a microtubule defect is the observation that stimulated $Opa1^{N\Delta}$ neutrophils also exhibit a failure in degranulation (Supplementary Fig. 3b, c), a process which is known to be dependent on microtubule assembly[32].

Therefore, we investigated the microtubule network in control $Lyz2^{Cre/Cre}$ and $Opa1^{N\Delta}$ neutrophils following activation and observed a disrupted microtubule network associated with OPA1 deficiency (Fig. 2a). Quantification of the microtubule network assembly demonstrated significant differences between $Opa1^{N\Delta}$ neutrophils compared to $Opa1^{flox/flox}$ and $Lyz2^{Cre/Cre}$ control neutrophils following GM-CSF/C5a stimulation (Supplementary Fig. 4a, b). Reduced microtubule network formation in $Opa1^{N\Delta}$ neutrophils was also observed following activation with GM-CSF/RNP ICs-Ab, GM-CSF/RNP ICs-SLE, PMA, or *P. aeruginosa* as compared to activated $Lyz2^{Cre/Cre}$ control neutrophils (Supplementary Fig. 4c). Inhibition of the microtubule network formation was also observed in human neutrophils expressing functionally inactive OPA1 (Fig. 2b). A quantitative analysis demonstrates that OPA1-defective human neutrophils are unable to assemble the microtubule network following combined GM-CSF/C5a stimulation (Supplementary Fig. 5a). To investigate the role of the microtubule network in DNA release, we used nocodazole and taxol, both known as highly efficient microtubule-disrupting agents[33]. As previously reported[19], both drugs completely prevented the formation of the microtubule network (Fig. 2c) as well as interdicting DNA release in stimulated normal human neutrophils (Fig. 2d). Taken together, these results indicate that OPA1 regulates microtubule network assembly which is required for the translocation of mitochondria, for degranulation, and for DNA release in activated mouse and human neutrophils.

On the other hand, $Opa1^{N\Delta}$ neutrophils exhibited normal production of reactive oxygen species (ROS) upon stimulation, excluding the possibility that OPA1 regulates the NADPH oxidase which is required for NET formation (Supplementary Fig. 4d)[2,34]. In addition, $Opa1^{N\Delta}$ neutrophils showed no abnormality with regard to actin polymerization upon activation (Supplementary Fig. 4e). Normal actin polymerization was also observed in human neutrophils expressing functionally inactive OPA1 (Supplementary Fig. 5b), confirming the findings obtained with primary mouse neutrophils. Summarizing, OPA1-deficient neutrophils are unable to form a microtubule network, but exhibit normal ROS production and actin polymerization under the same activation conditions.

**OPA1-deficient neutrophils exhibit decreased ATP production.** Super-resolution microscopy allowed us to clearly visualize a mitochondrial fragmentation in $Opa1^{N\Delta}$ neutrophils that was not seen in DNA-releasing control $Lyz2^{Cre/Cre}$ neutrophils in which mitochondria remained interconnected (Fig. 2a). Moreover, in contrast to normal human neutrophils that exhibit long filamentous mitochondria in the absence of stimulus, we also observed mitochondrial fragmentation in neutrophils of ADOA patients in both activated and resting conditions (Fig. 2b). These findings confirm previously published work suggesting a role for OPA1 in mitochondrial fusion[11].

It has been reported that the tightness of cristae junctions correlates with oligomerization of OPA1[10,11]. On the other hand, genetic depletion of OPA1 leads to disorganization of the cristae, resulting in problems with its assembly and with the stability of respiratory chain complexes and mitochondrial respiratory function[10,35]. To image mitochondria in $Opa1^{N\Delta}$ neutrophils, we used conventional transmission electron microscopy (TEM). While the number of mitochondria appeared to be independent of OPA1 expression (Fig. 3a, b), a finding which was supported by our mtDNA measurements (Fig. 3c), we also investigated mitochondrial transcription factor A (TFAM) expression as a measure of mtDNA levels. Equal levels of TFAM expression in control and $Opa1^{N\Delta}$ neutrophils were observed (Fig. 3d), also confirming indirectly that the mtDNA levels are unchanged in the absence of OPA1. Morphometric analysis revealed a diminished average mitochondrial length (Fig. 3a, e) and an increased width of the cristae (Fig. 3a, f) in $Opa1^{N\Delta}$ compared to $Lyz2^{Cre/Cre}$ neutrophils. Upon combined GM-CSF/C5a stimulation, the number of mitochondria increased owing to fission to a smaller average size in control neutrophils, while in $Opa1^{N\Delta}$ neutrophils, the number and size of mitochondria did not change significantly (Fig. 3a and Supplementary Fig. 6).

The increased width of the cristae as a consequence of OPA1 ablation is believed to be associated with an impaired respiration complex[35]. In addition, previously published work reported that muscle cells from adult ADOA patients displayed reduced ATP synthesis, suggesting dysregulation of mitochondrial metabolism as a consequence of OPA1 dysfunction[36]. Therefore, we analyzed the total cellular ATP level in $Opa1^{N\Delta}$ neutrophils and observed a reduction as compared to $Lyz2^{Cre/Cre}$ neutrophils (Fig. 3g).

**Complex I activity regulates neutrophil glycolysis and functions.** To test whether diminished ATP levels could contribute to the disruption of the microtubule network and the failure of DNA release, exogenous ATP or nicotinamide mononucleotide (NMN; to rescue $NAD^+$ levels, see below) was added to the cells 30 min before physiological activation of the neutrophils with GM-CSF/C5a. Indeed, addition of exogenous ATP or NMN were able to restore the ability of $Opa1^{N\Delta}$ neutrophils to form a normal

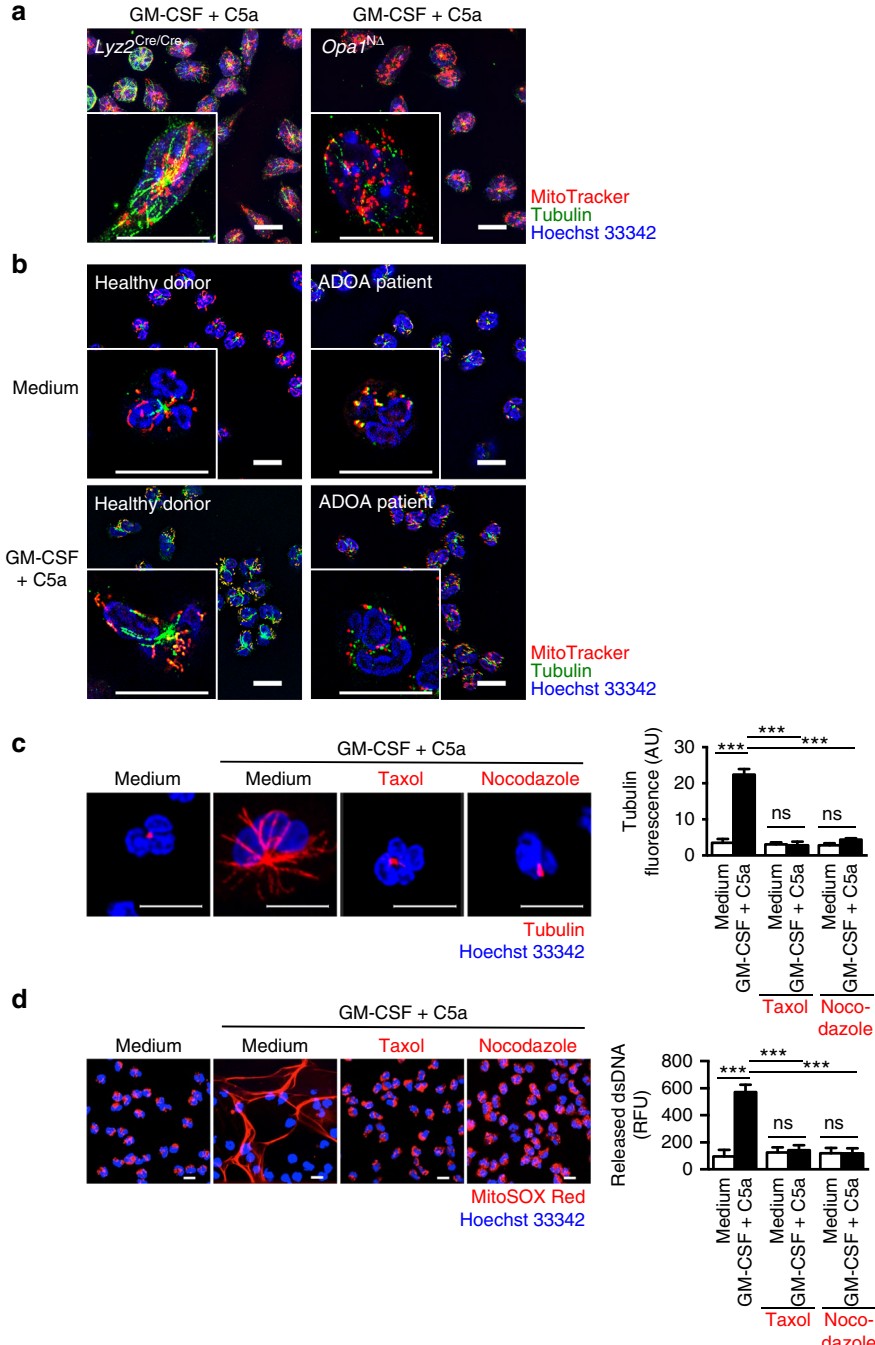

**Fig. 2** Absence of microtubule formation in OPA1-deficient human and mouse neutrophils. **a** Super-resolution microscopy. Microtubule formation and mitochondrial morphology of primary mature neutrophils from $Opa1^{N\Delta}$ and control mice following GM-CSF priming and subsequent C5a stimulation. Microtubules were stained with anti-α-tubulin antibody (green), the nucleus with Hoechst 33342 (blue) and mitochondria with MitoTracker® Orange (red). Images were acquired by ELYRA super-resolution microscopy. Bars, 10 μm. The data are representative of three independent experiments. Quantification and additional data including microtubule formation induced by other triggers are provided in Supplementary Fig. 4a–c. **b** Super-resolution microscopy. Microtubule formation and mitochondrial morphology in human blood neutrophils from control individuals and an ADAO patient (son, see Fig. 1) following GM-CSF priming and subsequent C5a stimulation. Microtubules were stained with anti-α-tubulin antibody (green), the nucleus with Hoechst 33342 (blue) and mitochondria with MitoTracker® Orange (red). Single cells are shown at higher magnification in the insets. Images were acquired by ELYRA super-resolution microscopy. Bars, 10 μm. The data are representative of three independent experiments. Quantification is provided in Supplementary Fig. 5a. **c** Confocal microscopy. Microtubule assembly was analyzed in human control neutrophils following pre-treatment with the indicated inhibitors and subsequent combined GM-CSF/C5a stimulation. Microtubules were stained with anti-α-tubulin antibody (red), the nucleus with Hoechst 33342 (blue). Bars, 10 μm. Right: Quantification of the microtubule network formation was performed by automated analysis of microscopy images using Imaris software. Values are means ± SEM. n.s., not significant; ***$p < 0.001$; $n = 5$. **d** Confocal microscopy. Human blood neutrophils from control individuals pretreated with the indicated inhibitors were stimulated with GM-CSF/C5a. Extracellular DNA was stained with MitoSOX™ Red and the nucleus with Hoechst 33342 (blue). Bars, 10 μm. Right: quantification of the released dsDNA in supernatants of activated neutrophils. Values are means ± SEM. n.s., not significant; ***$p < 0.001$; $n = 3$

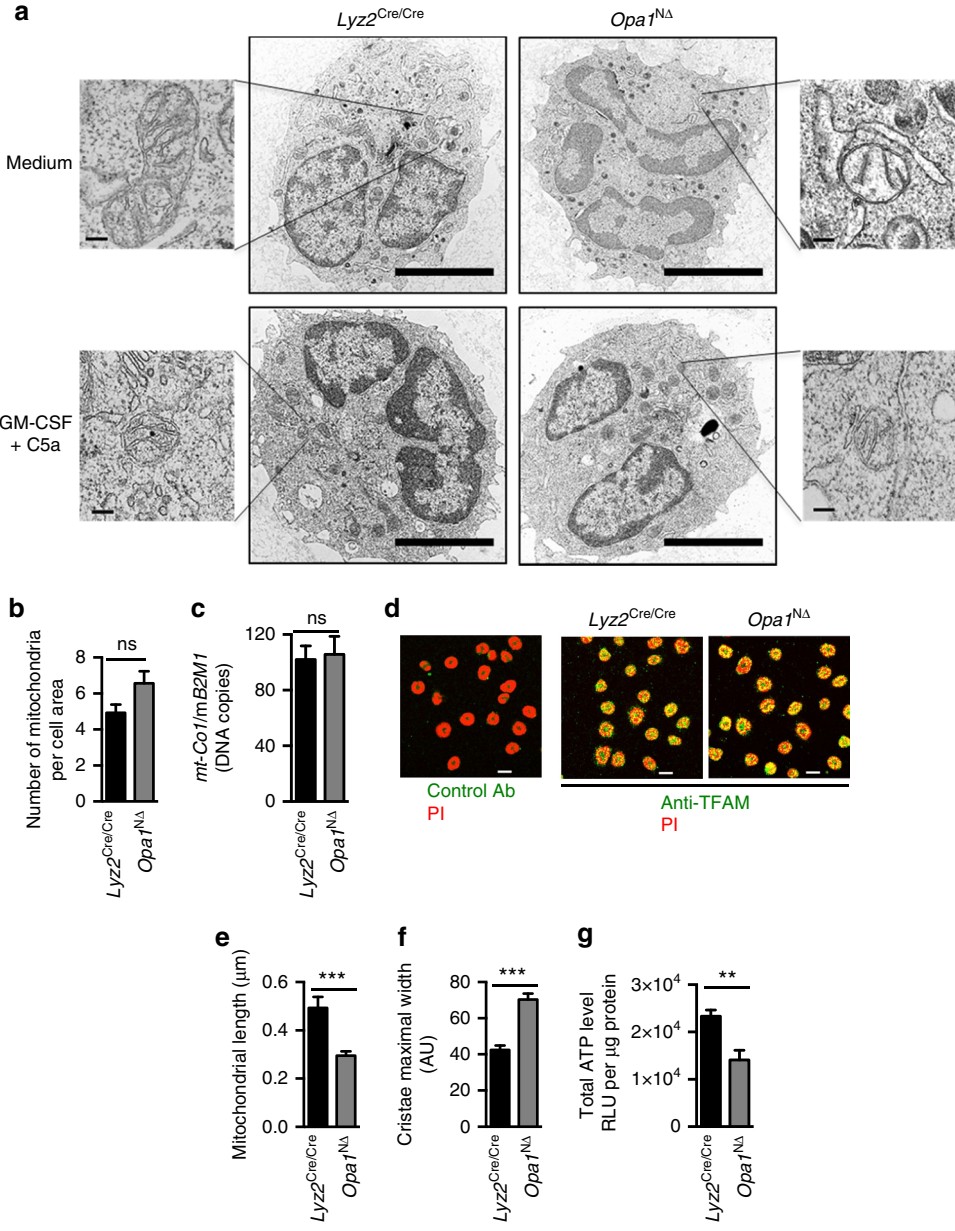

**Fig. 3** Lack of *Opa1* alters mitochondrial morphology and cristae structure. **a** Transmission electron microscopy (TEM). Primary mature neutrophils from *Opa1*$^{N\Delta}$ and control mice were fixed and analyzed. Representative images are shown. Bars, 5 µm. Insets to the sides, bars, 1 µm. Mitochondria were further analyzed at higher magnification and statistical analyses are provided below in (**b**), (**e**), and (**f**). **b** The average numbers of mitochondria per neutrophil was quantified in at least 100 cells. Values are means ± SEM ($n = 3$); n.s., not significant. **c** Quantitative PCR. Freshly purified mature neutrophils from *Opa1*$^{N\Delta}$ and control mice were analyzed for the number of DNA copies of mitochondrial mouse cytochrome *c* oxidase subunit 1 (*mt-Co1*) relative to mouse β2 microglobulin (*B2M*) which was used as a single copy nuclear-encoded reference gene. Values are means ± SEM ($n = 3$); n.s., not significant. **d** Confocal microscopy. Primary mature mouse neutrophils from *Opa1*$^{N\Delta}$ and control mice were stained for TFAM expression (green) and DNA (red) using anti-TFAM antibody and PI. Bars, 10 µm; $n = 3$. **e** Average mitochondrial major axis length in freshly purified mature neutrophils from *Opa1*$^{N\Delta}$ and control mice. Images were acquired by TEM and subsequently analyzed using the measurement points module of Imaris software. Data were collected from at least five mitochondria per cell and more than 50 neutrophils per experiment. Values are means ± SEM. ***$p < 0.001$; $n = 3$. **f** Morphometric analysis of cristae width in 60 randomly selected mitochondria of freshly purified mature neutrophils from *Opa1*$^{N\Delta}$ and control mice. Values are lengths in arbitrary units (AU) and shown as means ± SEM. ***$p < 0.001$; $n = 3$. **g** Total cellular ATP production by freshly purified mature neutrophils from *Opa1*$^{N\Delta}$ and control mice was measured by ATP-dependent luciferase activity using an ATP determination kit. Relative luciferase units (RLU) are shown as means ± SEM. **$p < 0.01$; $n = 5$. Quantification of the data obtained in activated neutrophils is provided in Supplementary Fig. 6

microtubule network (Fig. 4a) and to release extracellular DNA following activation (Fig. 4b). Quantification of dsDNA in neutrophil supernatants confirmed the DNA release by *Opa1*$^{N\Delta}$ neutrophils in the presence of exogenous ATP or NMN (Fig. 4b, right panel). Similar results for DNA release were obtained following stimulation of neutrophils with

PMA, *P. aeruginosa* or immune complexes (Supplementary Fig. 7a, b).

It should be noted that addition of exogenous ATP alone did not trigger DNA release either in human (Supplementary Fig. 8a) or in mouse neutrophils (Supplementary Fig. 8b), largely excluding the possibility that *Opa1*$^{N\Delta}$ neutrophils release DNA

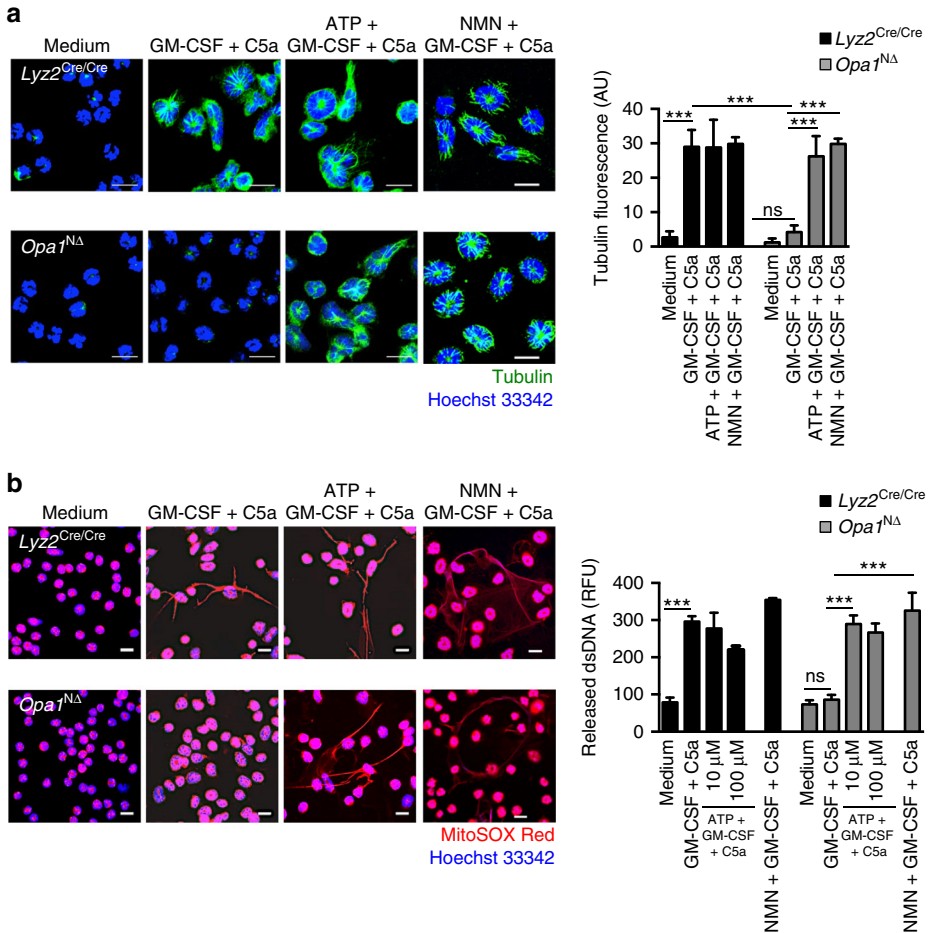

**Fig. 4** Exogenous ATP or NMN restore the microtubule network and the ability of activated $Opa1^{N\Delta}$ neutrophils to release DNA. **a** Confocal microscopy. GM-CSF primed and C5a-stimulated neutrophils of $Opa1^{N\Delta}$ and control mice were pretreated with 100 μM ATP or 500 μM NMN. Microtubules were stained with anti-α-tubulin antibody (green). Bars, 10 μm. Right: quantification of the microtubule network formation was performed using automated analysis of microscopy images with Imaris software. Values are means ± SEM. n.s., not significant; ***$p < 0.001$; $n = 5$. **b** Confocal microscopy. GM-CSF primed and C5a-stimulated neutrophils of $Opa1^{N\Delta}$ and control mice were pretreated with 10 and 100 μM ATP or 500 μM NMN. Extracellular DNA was stained with MitoSOX™ Red and the nucleus with Hoechst 33342 (blue). Bars, 10 μm. Right: quantification of released dsDNA in supernatants of activated neutrophils. Values are means ± SEM. n.s., not significant; ***$p < 0.001$; $n = 4$. Additional data including microtubule and NET formation induced by other stimuli known to trigger NETs are provided in Supplementary Figs. 7 and 8

as a consequence of P2X receptor activation. Moreover, the addition of exogenous ATP had no effect on the viability of either human (Supplementary Fig. 8c) or mouse neutrophils (Supplementary Fig. 8d). We also obtained no evidence for increased apoptosis of $Opa1^{N\Delta}$ compared to $Lyz2^{Cre/Cre}$ neutrophils (Supplementary Fig. 8d); a pan-caspase inhibitor failed to restore DNA release in GM-CSF/C5a-activated $Opa1^{N\Delta}$ neutrophils (Supplementary Fig. 8e), suggesting that, as an explanation for the inability of $Opa1^{N\Delta}$ neutrophils to form NETs, a difference in apoptosis regulation can be excluded.

In order to determine the source(s) for ATP production in human and mouse neutrophils, we applied specific inhibitors blocking different steps of mitochondrial respiration and glycolysis and measured total cellular ATP concentrations. After 70 min treatment of human neutrophils with the mitochondrial ATP synthase inhibitor oligomycin A or with the glycolysis inhibitor 2-deoxy-D-glucose (2-DG), we observed that inhibition of glycolysis clearly reduced ATP production, whereas inhibition of mitochondrial ATP synthase had no effect on neutrophil ATP levels (Fig. 5a). Moreover, we analyzed the ATP/ADP ratio as a measure for the energetic state of the cell and observed that the ratio was reduced by 2-DG in control $Lyz2^{Cre/Cre}$ mouse

neutrophils, but the reduced levels already attained in $Opa1^{N\Delta}$ neutrophils were not further reduced (Fig. 5b). Following GM-CSF/C5a stimulation, control neutrophils exhibit a reduced ATP/ADP ratio as a consequence of reduced ATP production, perhaps owing to the energy requirement for microtubule network rearrangements as previously suggested[37]. The fact that the fall in ATP concentration following neutrophil activation is not balanced by any measurable increase in ADP or AMP concentration has been reported previously[38]. Our data support the concept that glycolysis is the main source of ATP production in neutrophils[18] and indicate that $Opa1^{N\Delta}$ neutrophils exhibit a defect in glycolysis.

Despite the fact that glycolysis in the cytoplasm is the main source of ATP production in neutrophils, surprisingly, the mitochondrial complex I inhibitor rotenone also reduced ATP levels (Fig. 5a), indicating a link between glycolysis and the mitochondrial complex I pathway for ATP production in neutrophils. To examine whether Opa1 depletion has any effect on complex I activity in neutrophils, we analyzed the enzymatic activities of the mitochondrial complex I (CI) and complex III (CIII) normalized to citrate synthase (CS). The results clearly show that both resting and activated $Opa1^{N\Delta}$ neutrophils exhibit

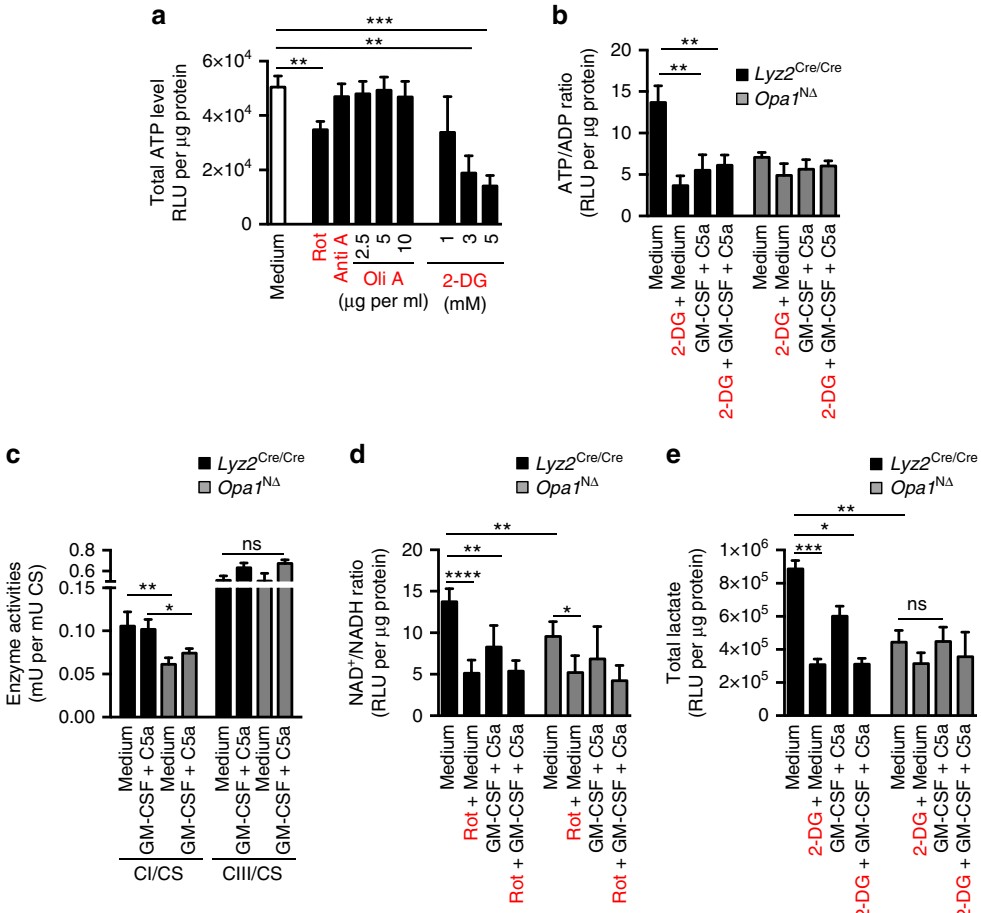

**Fig. 5** Reduced glycolytic ATP production in neutrophils of Opa1$^{N\Delta}$ mice is a consequence of reduced mitochondrial complex I activity. **a** ATP-dependent luciferase activity assay. Control human neutrophils were incubated with the indicated inhibitors for 70 min in a concentration-dependent manner. Relative luciferase unit (RLU) values are means ± SEM. **p < 0.01; ***p < 0.001; n = 3. **b** ATP/ADP bioluminescent assay. Neutrophils of Opa1$^{N\Delta}$ and control mice were pre-treated with 3 mM 2-DG for 70 min and the ATP/ADP ratio was measured in the presence and absence of GM-CSF/C5a. RLU values are means ± SEM. **p < 0.01; n = 6. **c** Complex I (CI), complex III (CIII) and citrate synthase (CS) activities of mitochondria isolated from resting and GM-CSF/ C5a-activated Opa1$^{N\Delta}$ and control Hoxb8 neutrophils were assessed in the presence and absence of 10 μM rotenone (Rot) using a spectrophotometer. Values are means ± SEM. n.s., not significant; *p < 0.05; **p < 0.01; n = 3. **d** NAD/NADH bioluminescent assay. NAD$^+$/NADH ratios in resting and GM-CSF/C5a-activated neutrophils of Opa1$^{N\Delta}$ and control mice were measured after 70 min incubation in the presence and absence of 10 μM Rot using a NAD/NADH determination kit. RLU values are means ± SEM. *p < 0.05; **p < 0.01; ****p < 0.0001; n = 5. **e** Lactate bioluminescent assay. Lactate levels in resting and GM-CSF/C5a activated neutrophils of Opa1$^{N\Delta}$ and control mice were measured after 70 min incubation in the presence and absence of 3 mM 2-DG using a lactate determination kit. RLU values are means ± SEM. n.s., not significant; *p < 0.05; **p < 0.01; ***p < 0.001; n = 4. ECAR and OCR data are provided in Supplementary Fig. 9

a significant defect in CI, but not CIII, activity (Fig. 5c), and in a diminished NAD$^+$/NADH ratio (Fig. 5d). In the process of glycolysis, glyceraldehyde-3-phosphate dehydrogenase (GAPDH) requires two NAD$^+$ molecules per glucose molecule. Therefore, NAD$^+$ availability is an important determinant for the rate of glycolysis[39,40]. Consistent with reduced glycolysis, we have also found reduced lactate levels in Opa1$^{N\Delta}$ as compared to Lyz2$^{Cre/Cre}$ neutrophils (Fig. 5e).

To better understand the basal level of glycolytic function in Opa1$^{N\Delta}$ and control Lyz2$^{Cre/Cre}$ mouse neutrophils, the extracellular acidification rate (ECAR; an indicator of aerobic glycolysis) and oxygen consumption rate (OCR; an indicator of OXPHOS) were analyzed. Consistent with the lactate findings (Fig. 5e), we found a general decrease in basal ECAR in Opa1$^{N\Delta}$ compared to control Lyz2$^{Cre/Cre}$ mouse neutrophils (Supplementary Fig. 9a, left panel). A lower ECAR in Opa1$^{N\Delta}$ neutrophils was also seen after blocking CI or CV activities or adding the mitochondrial uncoupling agent p-triflouromethoxyphenylhydrazone (FCCP)

(Supplementary Fig. 9a, right panel). Inhibition of CI activity had no effect in Opa1$^{N\Delta}$ neutrophils (Supplementary Fig. 9a, left panel), in agreement with the observation that these cells already exhibit a defect in CI activity (Fig. 5c). The addition of 2-DG abolished extracellular acidification in both Opa1$^{N\Delta}$ and control Lyz2$^{Cre/Cre}$ neutrophils (Supplementary Fig. 9a, left and right panels), confirming that neutrophils entirely depend on glycolysis. In agreement with this concept, both Opa1$^{N\Delta}$ and control Lyz2$^{Cre/Cre}$ neutrophils exhibit very low OCR (Supplementary Fig. 9b, left panel). The limited OCR in neutrophils is even further reduced in Opa1$^{N\Delta}$ neutrophils which, however, is an observation that gains statistical significance only at maximal respiration after adding FCCP (Supplementary Fig. 9b, right panel). The discrepancies between the clearly reduced CI activity and the only mildly affected OCR might be explained by the fact that mitochondria of neutrophils can receive electrons from glycolysis via glycerol-3-phosphate through the glycerophosphate dehydrogenase complex in the Q cycle[41]. Taken together, our data suggest that a lack of

OPA1 impairs complex I activity, reducing available $NAD^+$ levels, and hence the rate of glycolysis, resulting in reduced ATP production.

Since we observed a defect in the formation of microtubules and NETs following activation of OPA1-deficient human and mouse neutrophils, we investigated whether a pharmacological block in glycolysis and mitochondrial CI activity would also interdict the microtubule network assembly, thus resulting in inhibition of DNA release. Indeed, we found that blocking glycolysis (2-DG) or CI (rotenone or piericidin A), but not CIII (antimycin A) or ATP synthase (oligomycin A), resulted in an inability of normal mouse (Fig. 6a) and human neutrophils (Supplementary Fig. 10) to form microtubule networks or to release DNA upon combined activation with GM-CSF and C5a (Fig. 6b). To prove that ATP is sufficient to rescue the formation of the microtubule network and DNA release, we added either ATP or NMN to GM-CSF/C5a-activated control mouse neutrophils pre-treated with rotenone, piericidin A or 2-DG. Indeed, both exogenous ATP and NMN resulted in successful activation of neutrophils in spite of pharmacological block on CI activity or overall glycolysis (Fig. 6a, b; lower panels). These data support the concept that microtubule network assembly and DNA release depend on ATP generated by glycolysis, for which an OPA1-dependent mitochondrial complex I activity is required.

To demonstrate the importance of OPA1 for functional NET formation, we performed bacterial killing assays in vitro. We observed reduced bacterial killing of *P. aeruginosa* and *E. coli*–GFP by $Opa1^{N\Delta}$ as compared to $Lyz2^{Cre/Cre}$ neutrophils (Fig. 7a). Moreover, addition of DNase I lowered bacterial killing by $Lyz2^{Cre/Cre}$ neutrophils to levels similar to $Opa1^{N\Delta}$ neutrophils. Conversely, DNase I had no effect on $Opa1^{N\Delta}$ neutrophils, indicating that the lack of DNA release constitutes the principal cause of the reduced antibacterial activity of $Opa1^{N\Delta}$ neutrophils (Fig. 7a). To investigate the importance of mitochondrial complex I and complex III activities, we pre-treated $Lyz2^{Cre/Cre}$ neutrophils with rotenone and antimycin A. Rotenone, but not antimycin A, reduced the neutrophil killing activity, demonstrating the importance of complex I activity for functional NET formation. In contrast, rotenone had no effect on $Opa1^{N\Delta}$ neutrophils (Fig. 7a). Importantly, exogenous ATP or NMN rescued the defective bacterial killing activity of $Opa1^{N\Delta}$ neutrophils (Fig. 7a). To exclude the possibility that a failure of *P. aeruginosa* and *E. coli*–GFP killing by $Opa1^{N\Delta}$ neutrophils might be due to a lack of phagocytic activity, we performed phagocytosis assays using opsonized *E. coli*-GFP. Equal phagocytic activities were observed in $Opa1^{N\Delta}$ and $Lyz2^{Cre/Cre}$ neutrophils (Fig. 7b), suggesting that phagocytosis is largely independent of OPA1.

**$Opa1^{N\Delta}$ mice exhibit a reduced antibacterial defense capability**. NETs can bind and kill bacteria and fungi in the extracellular space[1,21]. To define the physiological importance of OPA1 in shaping the outcome of a bacterial lung infection, mice were intranasally inoculated with *P. aeruginosa*[42]. After 18 h, the median bacterial counts in the lungs of $Opa1^{N\Delta}$ mice were higher than for $Lyz2^{Cre/Cre}$ control mice (Fig. 7c). Moreover, systemic infection was more frequently observed in $Opa1^{N\Delta}$ mice, as shown by significantly higher median CFU counts in the spleen (Fig. 7d).

Neutrophil recruitment is a pivotal event of the response to *P. aeruginosa* infection[42,43]. To quantify neutrophil infiltration, lung homogenates were assessed for the neutrophil-specific enzyme myeloperoxidase activity (MPO). After 18 h infection, MPO enzymatic activity levels in lung homogenates of $Opa1^{N\Delta}$ mice were higher than in $Lyz2^{Cre/Cre}$ control mice, suggesting that

there is no defect in the neutrophil recruitment to the lungs of $Opa1^{N\Delta}$ mice (Fig. 7e).

To analyze the formation of NETs, lung tissues were investigated following staining of DNA and MPO. As expected, following infection, lung sections revealed the presence of NETs in $Lyz2^{Cre/Cre}$ control mice. In contrast, NET formation in $Opa1^{N\Delta}$ mice was largely abolished (Fig. 7f). Counting the numbers of infiltrating neutrophils revealed again evidence for increased neutrophil infiltration in the lungs in $Opa1^{N\Delta}$ compared to control mice ($3917 \pm 963$ versus $1742 \pm 319$ neutrophils per $mm^2$; $p < 0.05$). We additionally performed an automated analysis to quantify neutrophil numbers, confirming increased neutrophil infiltration in $Opa1^{N\Delta}$ compared to $Lyz2^{Cre/Cre}$ control mice (Fig. 7f, second right panel). Taken together, despite increased neutrophil recruitment to the site of infection and normal phagocytic activity, $Opa1^{N\Delta}$ mice fail to resolve bacterial lung infections, consistent with the lack of NET formation.

**Discussion**

Neutrophils are rapidly recruited in tissues during infections to eliminate pathogens[44], not only by phagocytosis, but also in the extracellular space by the formation of NETs[1]. Currently, there is a dispute in the literature regarding the source of extracellular DNA employed for the formation of NETs and whether there is a requirement for neutrophil death in NET formation[45]. Consequently, the molecular mechanism of NET formation remains unclear. Another matter of contradiction is the question of the functional significance of mitochondria in neutrophils. Since mitochondrial respiration is low in these cells, it has been suggested that neutrophils require mitochondria just for undergoing apoptosis[18]. On the other hand, it has been demonstrated that neutrophils possess a highly developed mitochondrial network that is important during infection and inflammation[46,47]. Here, we complement this body of evidence, extending a role of mitochondria in NET formation. Specifically, we report that mitochondrial complex I activity indirectly regulates glycolysis, which is required for microtubule network assembly and functional NET formation both in vitro and in vivo.

Recently, several researchers report release of mtDNA during NET formation, supporting our earlier findings[2]. For instance, mtDNA release by neutrophils has been intensely studied in systemic lupus erythematous (SLE) patients[48]. In this autoimmune disease, mtDNA acts as the most potent activator of plasmacytoid dendritic cells (pDCs) and the type I interferon (IFN) pathway[49]. Moreover, following activation of neutrophils with ribonucleoprotein immune complexes (RNP ICs), mitochondria were shown to be mobilized toward the cell surface where they released oxidized mtDNA in a mitochondrial ROS-dependent manner[29]. Additionally, a direct fusion of mitochondrial membranes with the plasmalemma as a potential mtDNA release mechanism in neutrophils has been postulated[50]. However, no conclusive molecular mechanism of mtDNA release has yet been reported.

To investigate the role of mitochondrial dynamics in NET formation, we have used super-resolution, confocal and TEM microscopy to analyze the interconnected tubular mitochondrial network in healthy human and mouse neutrophils. Down-regulation of different genes known to be responsible for mitochondrial fusion and fission using shRNA has pointed to an important role for OPA1 in facilitating DNA release. OPA1 is important for mitochondrial biogenesis and fusion[35]. Moreover, OPA1 regulates the shape of the mitochondrial cristae and their "tightness", a function needed for maintaining ion gradients, hence for overall respiratory efficiency[35]. Human and mouse neutrophils devoid of functional

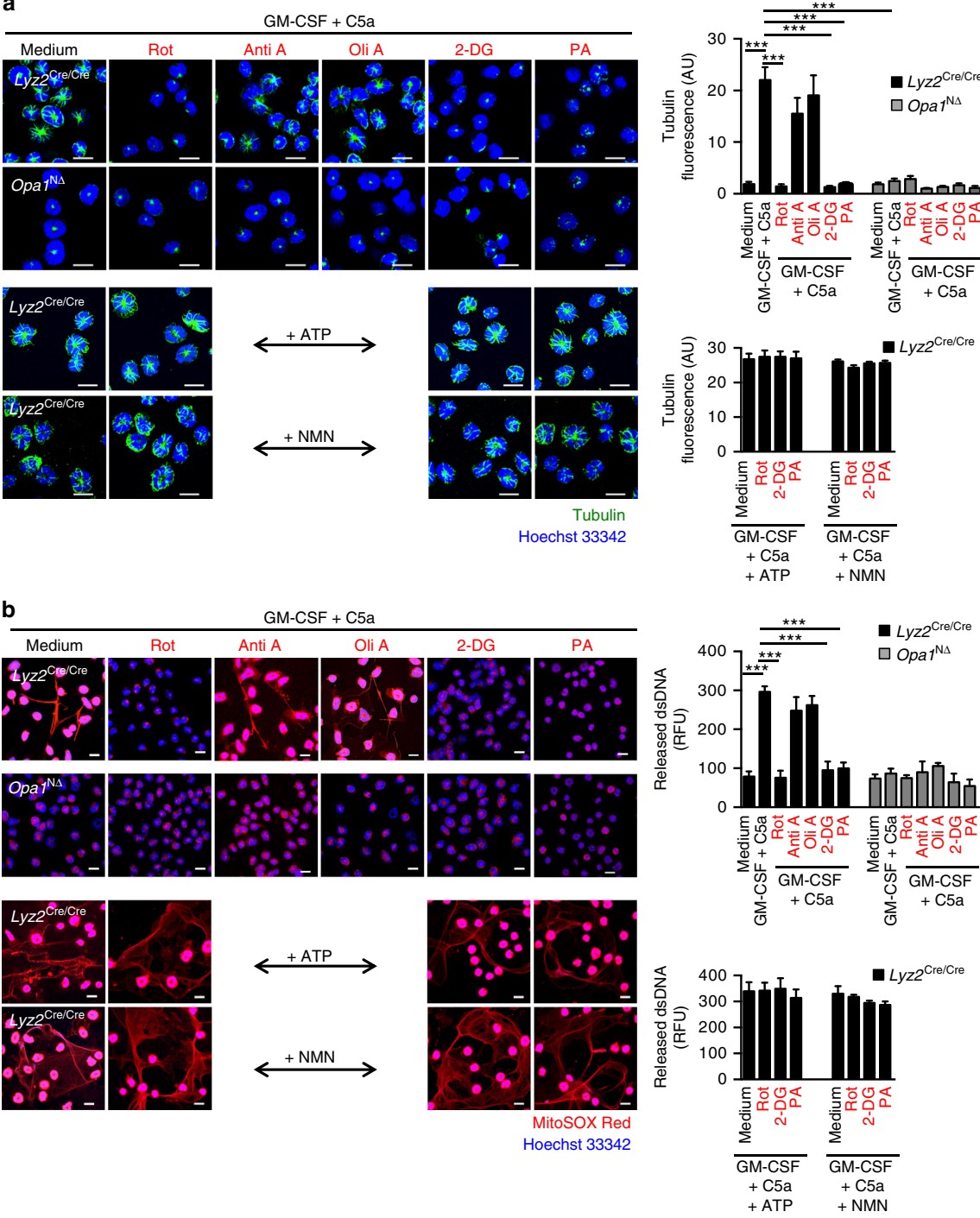

**Fig. 6** Pharmacological inhibition of mitochondrial complex I activity and glycolysis blocks microtubule network formation and extracellular DNA release. **a** Confocal microscopy. $Opa1^{N\Delta}$ and control neutrophils were pretreated with 10 μM rotenone (Rot), 5 μg per ml antimycin A (Anti A), 2.5 μg per ml oligomycin A (Oli A), 3 mM 2-DG, or 0.5 μM piericidin A (PA) for 30 min before GM-CSF priming and subsequent C5a activation. Lower panels: control mouse neutrophils were pretreated with 100 μM ATP or 500 μM NMN, incubated in presence or absence of the indicated inhibitors, and subsequently stimulated with GM-CSF/C5a. Microtubules were stained with anti-α-tubulin antibody (green). Bars, 10 μm. Right: Quantification of microtubule network formation was performed by automated analysis of microscopy images using Imaris software. Values are means ± SEM. ***$p < 0.001$; $n = 3$. **b** Confocal microscopy. Neutrophils of $Opa1^{N\Delta}$ and control mice were pretreated with 10 μM Rot, 5 μg per ml Anti A, 2.5 μg per ml Oli A, 3 mM 2-DG, or 0.5 μM PA for 30 min before GM-CSF priming and subsequent C5a activation. Lower panels: control mouse neutrophils were pretreated with 100 μM ATP or 500 μM NMN, incubated in presence or absence of the indicated inhibitors, and subsequently stimulated with GM-CSF/C5a. Extracellular DNA was stained with MitoSOX™ Red and the nucleus with Hoechst 33342 (blue). Bars, 10 μm. Right: quantification of released dsDNA in supernatants of activated neutrophils. Values are means ± SEM. ***$p < 0.001$; $n = 3$. Additional data using human neutrophils are provided in Supplementary Figs. 7 and 10

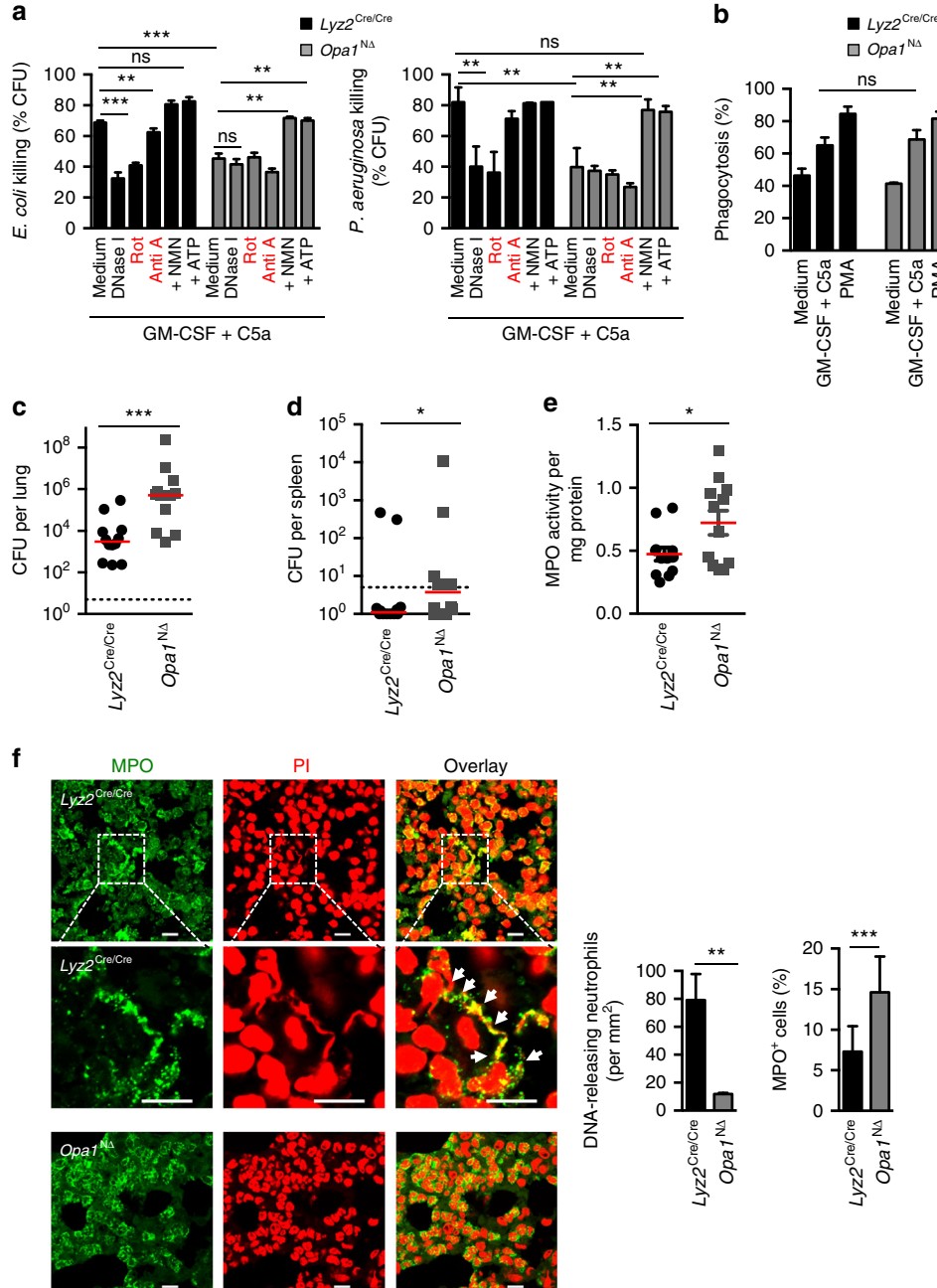

**Fig. 7** Anti-microbial response of $Opa1^{N\Delta}$ neutrophils and failure to clear *P. aeruginosa* lung infections. **a** Bacterial killing assays. Neutrophils of $Opa1^{N\Delta}$ and control mice were primed with GM-CSF and subsequently stimulated with C5a before co-culture with *E. coli*–GFP (left panel) and *P. aeruginosa* (right panel) in presence or absence of the indicated inhibitors or following 100 µM ATP or 500 µM NMN pre-treatment. Values are means ± SEM. n.s., not significant; **$p < 0.01$; ***$p < 0.001$; $n = 4$. **b** Phagocytosis assay. Phagocytosis of opsonized *E. coli*-GFP by $Opa1^{N\Delta}$ and control neutrophils was analyzed after 35 min. Values are means ± SEM ($n = 3$); n.s., not significant. **c, d** Bacterial clearance in vivo. $Opa1^{N\Delta}$ and control mice were intra-nasally inoculated with $2 \times 10^6$ CFUs of *P. aeruginosa*. Total CFU numbers of bacteria were assessed by plating homogenized lungs (**c**) and spleen (**d**) on agar plates. The medians are indicated and each symbol represents a value for an individual mouse. *$p < 0.05$; ***$p < 0.001$. **e** MPO enzymatic activity assay. Neutrophil infiltration of lungs in $Opa1^{N\Delta}$ and control mice was assessed by MPO activity in total lung homogenates. The medians are indicated and each symbol represents a value for an individual mouse. *$p < 0.05$. **f** Confocal microscopy. Lung sections of $Opa1^{N\Delta}$ and control mice were obtained after infection with *P. aeruginosa* and stained with anti-MPO antibody (green) and PI (red). Higher magnifications are shown in the middle panels; white arrows indicate co-localization of extracellular DNA and MPO. Bars, 10 µm. Right: quantification of the DNA-releasing and infiltrating neutrophils in the lungs. Quantification of MPO positive infiltrating neutrophils compared with total PI positive nuclei in the infected lungs using an automated slide scanner. Values are means ± SEM. **$p < 0.01$; ***$p < 0.001$; $n = 3$

OPA1 protein display not only a fragmented mitochondrial morphology, but also lack microtubule networks which are required for mitochondrial translocation, the process of degranulation, and the ability to form functional NETs upon activation. It should be noted that OPA1 cleavage in cerebellar granule neurons (CGNs) coincides

with extensive mitochondrial fragmentation and disruption of the microtubule network[31]. While we could pharmacologically demonstrate a requirement for the microtubule network for NET formation, we do not formally demonstrate why this is the case. However, disorientation of mitochondria in cells exhibiting a

disrupted microtubule array could explain the lack of extracellular DNA release by ADOA and $Opa1^{N\Delta}$ neutrophils.

Microtubule polymerization is an energy-dependent process[37]. It has been demonstrated that ATP is able to polymerize microtubules through an interaction at the exchangeable GTP site[51] and that reducing ATP synthesis leads to decreased microtubule stability[52]. We demonstrate that low levels of cellular ATP in OPA1-deficient neutrophils are insufficient for microtubule network and NET formation, a situation that could be reversed by addition of exogenous ATP or NMN which rescued $NAD^+$ levels. Surprisingly, however, actin polymerization and phagocytosis did not appear to be influenced by the lack of OPA1, suggesting that these events require less ATP. But why should OPA1, a mitochondrial protein known to be involved in mitochondrial ATP production, be important in neutrophils, which depend on glycolysis for ATP? Using neutrophils deficient in $Opa1$, we observed that a lack of OPA1 caused remodeling of mitochondrial cristae structure, resulting in reduced mitochondrial complex I activity and lower levels of $NAD^+$, which is a key participant in glycolysis[40]. Therefore, diminution of glycolysis as the main ATP production pathway in neutrophils is the reason for low cellular ATP levels in ADOA and $Opa1^{N\Delta}$ neutrophils, explaining their inability to form microtubule networks and functional NETs. These findings were confirmed with a pharmacological approach using inhibitors of mitochondrial complex I activity and glycolysis.

We also provide data showing that OPA1 function is important for anti-microbial defense by neutrophils in an in vivo model of $P.$ $aeruginosa$ lung infection. Mice lacking $Opa1$ in their neutrophils and macrophages did not form NETs and were less able to eliminate bacteria from the lungs. Moreover, these mice exhibited a greater probability of systemic spread of infection. Therefore, pharmacological activation of OPA1 in neutrophils might be a new option for fighting bacterial and fungal infections. Moreover, clinicians should be aware of the possibility that ADOA patients may exhibit an increased risk of infections. Taken together, the findings in this study provide further evidence for a role of mitochondria in the biology of neutrophils, particularly for NET formation, and extend the impact of an OPA1 function to the innate immune response.

## Methods

**Reagents**. Granulocyte/macrophage colony-stimulating factor (GM-CSF) was provided by Novartis Pharma (Nürnberg, Germany). Human and mouse C5a were purchased from Hycult Biotech (Uden, the Netherlands). MitoSOX Red, Alexa Fluor 488 Phalloidin, MitoTracker Orange CM-$H_2$TMRos, SYTO13, propidium iodide, CellMask Deep Red, Prolong Gold mounting media, Hoechst 33342, the Quant-iT$^{TM}$PicoGreen$^®$dsDNA Assay Kit and RPMI 1640/GlutaMAX medium were obtained from ThermoFisher Scientific (distributed by LuBioScience GmbH, Lucerne, Switzerland). X-VIVO 15 medium was from Lonza (Walkersville, MD, USA). Polyvalent human IgG was a kind gift from CSL Behring (Bern, Switzerland). Normal goat (Cat # NBP2-23475, dilution 1:400), rat (Cat # NBP2-33356, dilution 1:400), and rabbit (Cat # NBP1-71681, dilution 1:400) sera were purchased from Novus Biologicals, Abingdon, UK. ChromPure human IgG (Cat # 009-000-003, dilution 1:1000) and normal mouse sera (Cat # 015-000-120, dilution 1:400) were obtained from Milan Analytica AG (Rheinfelden, Switzerland). Bovine serum albumin (BSA), glutaraldehyde, saponin, sodium borohydride (NaBH$_4$), ATP, lipopolysaccharide (LPS, 055:B5), dihydrorhodamine 123 (DHR), 10% formalin solution, β-nicotinamide adenine dinucleotide (NADH), β-nicotinamide mononucleotide (NMN), piericidin A (PA), potassium cyanide (KCN), rotenone (Rot), antimycin A (Anti A), oligomycin A (Oli A), 2-deoxy-$D$-glucose (2-DG), nocodazole, taxol, propidium iodide, cytochalasin B, tris base, ubiquinone$_1$, bovine serum albumin (fatty acid free), sodium dodecyl sulfate (SDS), protease inhibitor cocktail, tryptic soy broth, and agar, Luria broth base (LB), $o$-dianisidine dihydrochloride, Triton X-100, $N$,$N$-dimethylformamide (DMF), and Nutridoma-SP were obtained from Sigma-Aldrich (Buchs, Switzerland). The NucleoSpin tissue kit for genomic DNA extraction was purchased from Macherey Nagel AG (Oensingen, Switzerland). IQ SYBR Green Supermix was from Bio-Rad Laboratories AG (Cressier, Switzerland). The Mitochondria Isolation Kit and Pierce BCA protein assay kit were from ThermoFisher Scientific as distributed by LuBioScience GmbH (Luzern, Switzerland). The CellTiter-Glo$^R$ luminescent assay, ADP/ATP-Glo$^™$

bioluminescent assay, and NAD/NADH-Glo$^™$ bioluminescent assay for detecting total oxidized and reduced nicotinamide adenine dinucleotides as well as the Lactate-Glo$^™$ bioluminescent assay were from Promega AG (Dübendorf, Switzerland). Deoxyribonuclease (DNase I) was from Worthington Biochemical Corporation (Lakewood, NJ, USA). Heparinized disposable glass pipettes were supplied by (Provet AG, Lyssach, Switzerland). The Mouse Neutrophil Enrichment Kit was from Stemcell Technologies (Grenoble, France). Pancoll-Human was purchased from PAN-Biotech GmbH (Aidenbach, Germany). 12-mm high-quality glass-cover slips for confocal microscopy analysis (Cat. # 1001/12) were purchased from Karl Hecht "Assistent" GmbH (Sondheim/Rhön, Germany).

**$Opa1^{N\Delta}$ mice**. We crossed mice bearing an $Opa1^{flox}$ allele, in which, following Cre-mediated recombination, the deletion of $Opa1$ exons 2 and 3 resulted in an aberrant exon1-exon4 transcript[35], with knock-in mice expressing Cre recombinase under the control of the lysozyme 2 endogenous promoter ($Lyz2^{Cre/Cre}$ mice). Lysozyme 2 is a marker of myeloid cells predominantly active in mature neutrophils and macrophages, and appears during late myeloid differentiation. The resulting $Opa1^{flox/flox}$_$Lyz2^{Cre/Cre}$ mice were designated $Opa1^{N\Delta}$ mice. $Lyz2^{Cre/Cre}$ mice or sometimes, additionally, $Opa1^{flox/flox}$ mice were used as controls in all experiments. Mouse genomic DNA was tested by PCR to verify the genotypes and homozygosity of $Opa1^{flox/flox}$ and $Lyz2^{Cre/Cre}$ alleles in $Opa1^{N\Delta}$ mice. All mouse experiments were approved by the Cantonal Veterinary Office of Bern, Switzerland (permission # BE5/15).

**Genomic PCR for mice deficient in $Opa1$**. For genotyping of mice, genomic DNA was extracted from an ear punch of mice using the NucleoSpin® Tissue Kit. PCR amplification was performed with 25 ng of total DNA using the following mouse primers ($Opa1$ forward: 5′-CAG TGT TGA TGA CAG CTC AG-3′; and reverse: 5′- CAT CAC ACA CTA GCT TAC ATT TGC-3′) and ($Lyz2$ forward: 5′-CCC AGA AAT GCC AGA TTA CG-3′, 5′-CTT GGG CTG CCA GAA TTT CTC-3′, and reverse: 5′-TTA CAG TCG GCC AGG CTG AC-3′), with SYBR Green Supermix. The samples were placed on a PCR thermo cycler amplified at 94 °C for 5 min, annealing at the required temperature for each primer for 30 s and 60 °C for 30 s with an extension at 72 °C for 1 min, for 40 cycles. The PCR-amplified products were separated on 2% polyacrylamide gels.

**cDNA analysis for exon 11 skipping in ADOA patients' neutrophils**. For $OPA1$ (NM_015560.2) exon 11 skipping, the following primer sequences were used: 3′-GTG CTG GAA AGA CTA GTG TGT TG -5′ and 3′-AGT ATG ATG GCA TTA GGA TTC TG-5′. Primers were synthesized by Microsynth AG (Balgach, Switzerland). Total RNA was extracted using the QIAGEN RNAeasy Kit. cDNA synthesis was performed using random hexamers with Invitrogen SuperScript II. The samples were placed on a PCR thermo cycler, amplified at 94 °C for 5 min, annealing at required temperature for each primer for 30 s and 60 °C for 30 s with an extension of 72 °C for 1 min, for 40 cycles. The PCR-amplified products were separated on 1.5% agarose gels.

**Isolation of mouse bone marrow and human blood neutrophils**. All mice were analyzed between 8 and 13 weeks of age. Blood was collected from the retro-orbital sinus of sedated mice, using heparinized disposable glass pipettes. Total leukocyte cell counts in blood were measured on a Scil Vet ABC Hematology Analyzer (Horiba Medical, Montpellier, France). For serum preparation, the blood was left to clot for 2 h at room temperature and centrifuged (20 min, 960 × $g$, 4 °C). Bone marrow cells were collected by flushing the femur with 5 ml of 2% FCS in PBS, using a 26-gauge needle, and filtering through a sterile 70-μm mesh nylon cell strainer. Bone marrow single-cell suspensions were then washed with medium and the cells counted with a hemocytometer, using the Türk's staining solution (Dr. Grogg Chemie AG, Stettlen, Switzerland). Primary bone marrow neutrophils were isolated by the negative selection technique, using an EasySep mouse neutrophil enrichment kit. Neutrophil purity was always higher than 90% as assessed by staining with the Hematocolor Set (Merck Millipore) followed by light microscopy analysis. Human neutrophils were isolated from healthy individuals and ADOA patients' blood[21]. Briefly, cell separation was performed by Ficoll-Hypaque centrifugation (Pancoll human from PAN$^{TM}$ BioTech). The granulocyte fraction was depleted of erythrocytes by lysis with a buffer containing $NH_4Cl$, $KHCO_3$ and EDTA. The resulting cell populations contained >95% neutrophils as assessed by Diff-Quik staining and light microscopy. Written, informed consent was obtained from all blood donors, and the Ethics Committee of the Canton Bern approved the study.

**Neutrophil activation**. Human and mouse neutrophils were seeded on 12-mm glass cover slips in X-VIVO$^{TM}$ 15 medium (Lonza, Walkersville, MD, USA), primed with 25 ng per ml GM-CSF for 20 min, and subsequently stimulated with $10^{-8}$ M C5a, 100 ng per ml LPS or co-cultured with $Escherichia$ ($E.$) $coli$ GFP M655 ($E.$ $coli$-GFP; a kind gift from E. Slack, ETH Zurich) for 15 min. In selected experiments, the following inhibitors were used 30 min before GM-CSF priming: Oligomycin A (2.5, 5, or 10 μg per ml), 2-DG (1, 3, 5 mM), rotenone (10 μM), antimycin A (5 μg per ml), Q-VD (20 μM), taxol (1 μM) and nocodazole (5 μM). In other experiments, cells were pre-cultured for 30 min in the presence of ATP (10,

100 µM, or 1 mM) or NMN (500 µM) and then stimulated for 45 min, besides GM-CSF/C5a, also with RNP-ICs-SLE [IgG, purified from with SLE antibody positive human plasma (Cat # DA1805, Trina Bioreactive AG, Nänikon, Switzerland, dilution 1:33), mixed with small nuclear ribonucleoprotein (SmRNP) (Cat # ATR01-02, Arotec, New Zealand, dilution 1:100)] or RNP-ICs-Ab [an anti-damaged DNA/RNA monoclonal antibody (clone 15A3, Cat # SMC-155, Stress-Marq Biosciences, Victoria, Canada, dilution 1:33) mixed with SmRNP][29] or *P. aeruginosa* (ATCC-BAA-47; strain HER-1018)[42]; with a multiplicity of infection of 10:1 (bacteria to neutrophils).

**PLB-985 cells**. The human myeloid leukemia cell line PLB-985 originally from Collection of Microorganisms and Cell Cultures (DSMZ, Germany Cat # ACC 139), and was kindly provided by V. Witko-Sarsat, Cochin Institute, Paris, France, was cultured in RPMI 1640+ GlutaMAX medium supplemented with 10% FCS, 50 units per ml penicillin and 50 µg per ml streptomycin. Cell cultures were passaged twice or three times a week to maintain a cell density between $2 \times 10^5$ and $1 \times 10^6$ per ml. For granulocytic differentiation, PLB-985 cells were cultured at a density of $2 \times 10^5$ per ml in RPMI 1640 medium supplemented with 0.5% N,N-dimethyl-formamide (DMF), 1% Nutridoma-SP and 0.5% FCS[53]. The medium was changed once on day 3 during the differentiation period. On day 6, the cells were centrifuged and resuspended in fresh medium.

**Down-regulation of mitochondria-shaping proteins by shRNA**. Human and mouse short hairpin RNAs (shRNA) were subcloned in HIV-derived lentivirus vector pLKO.1-puromycin (human cells) or G418 (mouse cells) with a U6 promoter that had been purchased from Sigma-Aldrich. Lentiviral vector containing shRNA were calcium phosphate transfected together with a lentivirus envelope protein vector pMD2G and the packaging vector psPAX2 (provided by D. Trono, École Polytechnique Fédérale de Lausanne, Lausanne, Switzerland) into 293T/17 cells from American Type Culture Collection (ATCC, USA Cat # CRL-11268). Supernatants were collected after 24 and 48 h, cleared by low-speed centrifugation ($425 \times g$ for 10 min), filtered through a 0.22-µm filter (Merck Millipore), and the lentivirus vectors encoding human and mouse shRNAs added to human myeloid leukemia cell line (PLB-985) and mouse Hoxb8 cell lines[20], respectively. pLKO.1 constructs encoding shRNAs against human and mouse were obtained from Sigma-Aldrich (Mission shRNA).

shRNA primer sequences: *DRP1* (TRCN0000001097, CCG GGC TAC TTT ACT CCA ACT TAT TCT CGA GAA TAA GTT GGA GTA AAG TAG CTT TTT), mouse *Drp1* (TRCN0000012605, CCG GGC TTC AGA TCA GAG AAC TTA TCT CGA GAT AAG TTC TCT GAT CTG AAG CTT TTT), human *FIS1* (TRCN0000155375, CCG GCA AGA GCA CGC AGT TTG AGT ACT CGA GTA CTC AAA CTG CGT GCT CTT GTT TTT TG), mouse *Fis1* (TRCN0000124386, CCG GCC TGG TTC GAA GCA AAT ACA ACT CGA GTT GTA TTT GCT TCG AAC CAG GTT TTT G), human *MFN1* (TRCN0000051833, CCG GGC TCC CAT TAT GAT TCC AAT ACT CGA GTA TTG GAA TCA TAA TGG GAG CTT TTT G), mouse *Mfn1* (TRCN0000081401, CCG GGC GTT TAA GCA GCA GTT TGT ACT CGA GTA CAA ACT GCT GCT TAA ACG CTT TTT G), human *OPA1* (TRCN0000082846, CCG GCC GGA CCT TAG TGA ATA TAA ACT CGA GTT TAT ATT CAC TAA GGT CCG GTT TTT G), or mouse *Opa1* (TRCN0000091111, CCG GCC GAC ACA AAG GAA ACT ATT TCT CGA GAA ATA GTT TCC TTT GTG TCG GTT TTT G).

The down-regulation of DRP1, FIS1, MFN1, and OPA1 after antibiotic selection was controlled by qPCR using primers as follows: human *DRP1* (forward: 5′-TTC AAT CCG TGA TGA GTA TGC-3′, reverse 5′-TTA GAA GAG ACT GAT ACT GAG CA-3′), mouse *Drp1* (forward: 5′-GAG TGT AAC TGA TTC AAT-3′, reverse 5′-GCA TAA GTA ACC TAT TCA-3′), human *FIS1* (forward: 5′-AAG AGC ACG CAG TTT GAG-3′, reverse 5′-CAG GTA GAA GAC GTA ATC CC-3′), mouse *Fis1* (forward: 5′-AGC GGG ACT ATG TCT TCT-3′, reverse 5′-GCA CAT ACT TTA GAG CCT TT-3′), human *MFN1* (forward: 5′-GTT GGA GCG GAG ACT TAG-3′, reverse 5′-CGG ATT CTT ATA TGT TGC TTC AA-3′), mouse *Mfn1* (forward: 5′-TAA TGG CAG AAA CGG TAT-3′, reverse 5′-TTC CTG TAT GTT GCT TCA-3′), or human *OPA1* (forward: 5′-GGT TGT TGT GGT TGG AGA T-3′, reverse 5′-AGA GTC ACC TTA ACT GGA GAA-3′), mouse *Opa1* (forward: 5′-AAG TGA CAA GCA TTA CAG G-3′, reverse 5′-CTC CAA GAT CCT CTG ATA CT-3′).

**Reactive oxygen species measurements**. ROS measurements in human and mouse neutrophils were performed using fluorescent detection of ROS activity by flow cytometer[21]. Neutrophils ($2 \times 10^6$ per ml) were resuspended in X-VIVO™ 15 medium and neutrophils primed with GM-CSF and subsequently stimulated with C5a. For positive control experiments, we stimulated neutrophils with PMA (25 nM) in the absence of GM-CSF priming. DHR 123 was added to the cells at the final concentration of 1 µM. The reaction was stopped by adding 200 µl of ice-cold PBS, and the ROS activity of the samples was immediately measured by flow cytometry (BD FACSCalibur) and quantified using FlowJo software (Tree Star, Ashland, OR, USA).

**NET formation**. Neutrophils were fixed with 4% paraformaldehyde for 10 min, washed with PBS, pH 7.4, and permeabilized using 0.05% saponin in PBS pH 7.4

for 2.5 min at room temperature. Subsequently, staining was performed in the presence of 0.05% saponin. Nonspecific binding was reduced by pre-incubation of the adherent cells with a blocking buffer (containing human immunoglobulins) at room temperature for 20 min. Indirect immunofluorescence staining was performed, using polyclonal goat anti-mouse neutrophil elastase (M-18, Cat # SC-9518, Santa Cruz Biotechnology, dilution 1:50) or polyclonal goat anti-mouse MPO (E-15, Cat # SC-34159, Santa Cruz Biotechnology, dilution 1:50) and rabbit anti-TFAM (Cat # NBP1-67995, Novus Biologicals, dilution 1:100) primary antibodies, diluted in blocking solution, and incubated overnight at 4 °C for staining tissue sections or 2 h for isolated neutrophils[21]. As staining controls, either mouse or rabbit control antibodies at the corresponding concentrations were employed. Appropriate Alexa Fluor® 488 (Cat # A11055, ThermoFischer Scientific, dilution 1:400) or Cyanine Cy™3 (Cat # 715-167-003, Jackson ImmunoResearch, dilution 1:100)-conjugated secondary antibodies were incubated at room temperature for 1 h. DNA was stained with PI (10 µg per ml) or monoclonal mouse anti-dsDNA (clone AE-2, Cat # MAB1293, Merck Millipore, dilution 1:200). Staining with cell-permeable fluorescent dyes, such as MitoSox Red (5 µM) or MitoTracker Orange (1 µM), was performed in live cells prior to fixation according to the corresponding manufacturers' instructions. Samples were mounted in Prolong Gold mounting medium, image acquisition was performed using the confocal laser scanning microscope LSM 700 (Carl Zeiss Micro Imaging, Jena, Germany) with a ×63/1.40 oil DIC objective and analyzed with Imaris software (Bitplane AG, Zurich, Switzerland) as previously reported[7,21].

**Super-resolution multicolor imaging**. High-resolution imaging was performed with a structured illumination microscope Elyra S.1 (Carl Zeiss) equipped with a Plan-Apochromat ×63/1.4 NA objective and an iXon 885 EM-CCD camera (Andor Technology), and was piloted with the ZEN Blue 2010 D software (Carl Zeiss). Image reconstruction was carried out with the Zen software (Carl Zeiss).

**Quantification of released dsDNA in culture supernatants**. Neutrophils ($2 \times 10^6$ neutrophils per 500 µl of X-VIVO™ 15 medium) were stimulated as described above. At the end of the incubation period, a low concentration of DNase I (2.5 U per ml; Worthington Biochemical Corporation, Lakewood, NJ, USA) was added for an additional 10 min. Reactions were stopped by addition of 2.5 mM EDTA, pH 8.0. Cells were centrifuged at $200 \times g$ for 5 min. One hundred microliters supernatant were then transferred to black, glass-bottom 96-well plates (Greiner Bio-One GmbH) and the fluorescent activity of PicoGreen dye bound to dsDNA was excited at 502 nm. The fluorescence emission intensity was measured at 523 nm using a spectrofluorimeter (SpectraMax M2, Molecular Devices, Biberach an der Riß, Germany), according to the instructions described in the Quant-iT™PicoGreen® assay kit[21].

**Microtubule formation**. Neutrophils were seeded on glass coverslips and stimulated as described above. Samples were fixed in mixture of 4% formaldehyde, and 0.1% glutaraldehyde in PBS (pH 7.4), and permeabilized first with 0.5% Triton X-100 for 15 min, followed by a 15-min incubation with 0.5% sodium dodecyl sulphate (SDS) at room temperature. Autofluorescence was then quenched with 0.5 mg per ml sodium borohydride (NaBH$_4$) in PBS (pH 7.4) for 10 min on ice. Nonspecific binding to the Fc-receptor was blocked with a blocking buffer [7.5% BSA, 30% human IgG polyvalent (IVIG), and 30% normal goat sera (matching species with secondary antibody) and 1% ChromPure human IgG in PBS] at room temperature for 1 h. Anti-α-tubulin monoclonal antibody (clone DM1A, Cat # T9026, Sigma-Aldrich, dilution 1:100,000) was diluted in blocking buffer and incubated overnight at 4 °C, subsequently washed three times in PBS (pH 7.4) with 0.025% Triton X-100, and then further incubated with the secondary antibody, Alexa-555 or -488 goat anti-mouse IgG (Cat # A32727 or A32723, ThermoFischer Scientific, dilution 1:1000) diluted in 7.5% BSA plus 0.025% Triton X-100 at room temperature for 1 h. Following extensive washing with PBS (pH 7.4) plus 0.025% Triton X-100, the samples were counterstained with 1 µg per ml Hoechst 33342 at room temperature for 5 min in the dark. After several washes, the samples were mounted on glass slides using Prolong Gold mounting medium[19,54]. Images were acquired using the LSM 700. Ten representative pictures from each condition, containing more than 20 cells, were subjected to automated analysis to measure fluorescence mean intensity of anti-α-tubulin before and after the activation period using automated isosurface module of Imaris software.

**F-actin staining and quantification**. Neutrophils were seeded on 12-mm glass cover slips and treated as described above. Staining with MitoTracker Orange (1 µM), was performed in live cells prior to fixation. Cells were fixed with 4% paraformaldehyde for 5 min, subsequently washed three times in PBS (pH 7.4), and permeabilized with 0.5% saponin in buffer A (1% BSA in PBS) for 5 min at room temperature. F-actin was stained using 1.25% solution (7 µM) of Alexa Fluor® 488 Phalloidin (green) in PBS (pH 7.4) plus 0.05% BSA and 0.5% of saponin for 15 min at room temperature, protected from light. Cells were washed two times in PBS (pH 7.4) and counterstained with 1 µg per ml Hoechst 33342 to visualize the nuclei. After washing three times in PBS (pH 7.4), samples were mounted in Prolong Gold mounting media. Slides were examined and images acquired by confocal laser scanning microscopy (LSM 700). Ten representative pictures from each condition,

containing more than 20 cells, were subjected to automated analysis to measure fluorescence mean intensity of F-actin before and after the activation period, using the automated isosurface module of Imaris software.

**Analysis of mitochondrial localization.** Imaris software was employed for quantification of mitochondrial localization. MitoTracker-stained mitochondria were imaged with an LSM 700 confocal microscope and processed using measurement point module and tab Intensity tool within Imaris software to quantitatively analyze images. Ten representative pictures from each condition, containing more than 20 cells were used to determine the localization of mitochondria with respect to their distance from the plasma membrane[55].

**Immunoblotting.** Cell lysates of primary bone marrow-derived mouse or human blood neutrophils were prepared by lysing the cells with lysis buffer containing 50 mM Tris pH 7.4, 150 mM NaCl, 10% glycerol, 1% Triton X-100, 2 mM EDTA, 10 mM sodium pyrophosphate, 50 mM sodium fluoride, 200 µM Na3VO4 and 1 mM PMSF in H₂O. Shortly before use, a protease inhibitor cocktail was added to the lysis buffer. Cells were lysed for 25 min on ice, with frequent vortexing. After the lysis, supernatants were collected following high-speed centrifugation (10 min, $17,949 \times g$, 4 °C). In all, 30–50 µg of protein was loaded on 12% SDS polyacrylamide gels (Expedeon, Lucerna-Chem). The samples were separated by electrophoresis under reducing conditions and transferred onto polyvinylidene difluoride (PVDF) membranes (Immobilon-P; Merck Millipore). Membranes were blocked with 5% non-fat dry milk in Tris-buffered saline solution containing 0.1% Tween 20 (TBST) for 1 h, followed by incubation with monoclonal anti-OPA1 antibody (clone 18/OPA1, Cat # 612606, BD Biosciences, dilution 1:1000) in TBST and 5% non-fat dry milk at 4 °C overnight. Membranes were washed using TBST, then incubated with the corresponding HRP-conjugated secondary antibody (Cat # GENA931, Sigma-Aldrich, dilution 1:10,000) and visualized by chemiluminescence (ECL Western Blotting Analysis System; GE Healthcare or Luminata Forte; Merck Millipore).

**Phagocytosis assay.** E. coli-GFP ($25 \times 10^6$ per ml) were opsonized with 2% mouse serum in 1× Hank's Balanced Salt Solution (HBSS; LuBioScience GmbH) for 15 min (rotating end-over-end, 37 °C). Primary bone marrow-derived mouse neutrophils ($0.5 \times 10^6$) were resuspended in 200 µl of X-VIVO™ medium without phenol red or antibiotics and 200 µl of opsonized bacteria was then added to the cells. The cells were incubated with the bacteria for 35 min (rotating end-over-end, 37 °C). Phagocytosis was stopped by adding 400 µl of ice-cold PBS with 0.02% EDTA to the cells. After one washing step with 400 µl of ice-cold PBS, the cells were analyzed by flow cytometer (BD FACSCalibur).

**Bacterial killing assay.** A single colony of E. coli-GFP was cultured in Luria broth base (LB) medium (Sigma-Aldrich) or a colony of P. aeruginosa was cultured in tryptic soy broth at 37 °C, shaking, overnight. The bacterial cultures were diluted 1:100 in LB medium, grown to mid-logarithmic growth phase ($OD_{600} = 0.7$) and centrifuged at $1000 \times g$ for 5 min. Bacterial pellets were washed twice with 1× Hank's Balanced Salt Solution (HBSS; LuBioScience GmbH) and again gently centrifuged at $100 \times g$ for 5 min to remove any clumped bacteria. Bacteria were opsonized with 10% mouse sera (heat-inactivated) in 1× HBSS, rotating end-over-end at 37 °C for 20 min, and then used immediately. Activated neutrophils ($1 \times 10^7$ per ml in RPMI 1640 plus 2% FCS in absence of antibiotics) were mixed with an equal volume of opsonized bacteria ($5 \times 10^7$ per ml) at 1:5 ratios, in the presence or absence of 100 U per ml DNase I (Worthington Biochemical Corporation, Lakewood, NJ, USA). The co-cultures of cells and bacteria were rotated end-over-end at 37 °C for 30 min. At the end of the incubation period, an equal volume of ice-cold 1× HBSS was added to each tube to stop the reaction, and cells were pelleted by gentle centrifugation (5 min, $100 \times g$, 4 °C) using a swing-out rotor. Supernatants containing bacteria were collected, diluted 1:300, plated on agar, and grown overnight before counting the colonies. The tubes containing bacteria alone were treated the same way and used as controls[21].

**Inoculum preparation.** For each experiment, P. aeruginosa inoculum was freshly prepared from a frozen glycerol stock and grown overnight on tryptic soy agar (TSA)[56]. The inoculum was washed in PBS, and the concentration was adjusted by spectrophotometry. The actual CFU in the inoculum was determined each time by plating serial dilutions on TSA. Mice were inoculated with $2 \times 10^6$ CFU per mouse.

**In vivo lung infection.** Groups of mice were sedated with 100 mg per kg ketamine and 10 mg per kg xylazine and intranasally inoculated by applying 10 µl of inoculum onto each nare. Mice were killed after 18 h for tissue harvest. The dissected spleen and right lung were homogenized, serially diluted in 1% protease peptone, and plated in duplicate on TSA. Aliquots of lung homogenates were stored at −80 °C until further analysis. Left lungs were fixed in formalin solution and processed for immunofluorescence staining. Lung sections were stained using goat anti-mouse MPO antibody (E-15, Cat # SC-34159, Santa Cruz Biotechnology, dilution 1:50) at 4 °C overnight, followed by conjugated Alexa Fluor 488 donkey

anti-goat IgG (Cat # A11055, ThermoFischer Scientific, dilution 1:100) at room temperature for 1 h, and subsequently with 10 µg per ml PI for 10 min.

**MPO activity assay.** MPO activity was measured in triplicate using a spectrophotometric assay in 50 mM potassium phosphate, pH 6, containing 0.167 mg per ml o-dianisidine dihydrochloride and 0.0005% hydrogen peroxide.

**Measurement of mitochondrial DNA copy number.** Total genomic DNA was extracted from primary mouse or human neutrophils with a NucleoSpin® tissue kit. Quantitative PCR (qPCR) assays were performed using iQ SYBR Green Supermix (Bio-Rad Laboratories). Mouse primers[57] were as follows: Mouse cytochrome c oxidase subunit 1 (mt-Co1) (forward: 5′-CCA GTG CTA GCC GCA GGC AT-3′, reverse 5′- TCT GGG TGC CCA AAG AAT CAG AAC A-3′) and for nDNA primers mouse β2 microglobulin (mB2M1) (forward: 5′- ATG GGA AGC CGA ACA TAC TG-3′, reverse 5′-CAG TCT CAG TGG GGG TGA AT-3′). Human primers[25] were as follows: human D-Loop (forward: 5′-CAT CTG GTT CCT ACT TCA GGG-3′, reverse 5′-CCG TGA GTG GTT AAT AGG GTG-3′) and human ATP synthase protein 8 (MT-ATP8) (forward: 5′-ATG GCC CAC CAT AAT TAC CC-3′, reverse 5′-CAT TTT GGT TCT CAG GGT TTG-3′), and human β2 microglobulin (B2M) (forward: 5′-TGC TGT CTC CAT GTT TGA TGT ATC T-3′, reverse 5′-TCT CTG CTC CCC ACC TCT AAG T-3′). Primers were synthesized by Microsynth AG. Duplicate real-time PCRs were performed in the iQ5 Multicolor Real-time PCR Detection System (Bio-Rad Laboratories), with 25 ng extracted genomic DNA, and 200 nM of each primer in 25 µl final PCR mix. The cycling variables were 3 min at 95 °C followed by 40 cycles of 10 s at 95 °C and 30 s at 55 °C.

**Enzyme assays of mitochondrial respiratory chain complexes I and III.** Mitochondria from differentiated Hoxb8 mouse neutrophils were prepared using the Mitochondria Isolation Kit according to manufacturer's protocol (Thermo Fisher Scientific). Isolated mitochondria were resuspended in 10 mM ice-cold hypotonic Tris buffer (pH 7.6) and stored at −80 °C. Before use, mitochondria were subjected to three cycles of sonication to disrupt the mitochondrial membrane. The enzymatic activities of CI, CIII and citrate synthase were determined spectrophotometrically as described[58] in 10–20 µg extracted protein using 1 ml sample cuvettes at 30 °C. CI activity was measured as the rotenone-sensitive NADH: decylubiquinone oxidoreductase reaction and the decrease in absorbance of NADH oxidation was read at 340 nm. CIII function was measured as the antimycin A-sensitive decylubiquinol:cytochrome c reductase reaction and increasing reduction of cytochrome c was read at 550 nm. Citrate synthase was measured as the rate of production of coenzyme A from oxaloacetate by measuring the reduction of 5,5′-dithiobis-2-nitrobenzoic acid at 412 nm. The values were normalized to the amount of protein in each sample. Enzymatic CI and CIII activities were calculated as ratios to the mitochondrial marker enzyme citrate synthase (mU per mU citrate synthase).

**Extracellular flux analysis using Seahorse XF24.** OCR and ECAR were investigated simultaneously using a Seahorse XF24 bioanalyzer (Agilent Technologies, CA, USA). Primary mouse neutrophils were seeded in unbuffered culture medium at 480,000 per well in polylysine-coated XF24-well assay plates. After the initial assessment of basal OCR and ECAR rates, sequential exposures to modulators of mitochondrial function were injected at optimized concentrations using a standard mitochondrial stress test paradigm. First, mitochondrial phosphorylation was inhibited by adding oligomycin (2 µM). Next, the mitochondrial electron transport chain was stimulated maximally by the addition of the uncoupler carbonyl cyanide-p-trifluoromethoxyphenylhydrazone (1.5 µM, FCCP). Finally, mitochondrial respiration was inhibited at the level of complex I by addition of rotenone (1 µM), and glycolysis inhibited by injection of 2-deoxy-D-glucose (50 mM, 2-DG). Steady-states were calculated as the mean of three OCR measurement cycles. After each experiment, OCR and ECAR data were normalized to corresponding cell numbers in each well using protein determination.

**ATP/ADP, NAD⁺/NADH, and lactate measurements.** Resting and activated mouse neutrophils ($4 \times 10^6$ per ml) were treated with and without glycolysis (2-DG) or CI (rotenone) inhibitors for 70 min at 37 °C in DMEM medium (no glucose, no glutamine, no phenol red (Cat # A14430-01, Thermo Fisher Scientific). 12 mM D-glucose was added for lactate and ATP/ADP measurements. For NAD⁺/NADH quantitative analysis, cells were treated as described above, but cultured in PBS. Upon activation, cells were placed on ice for lactate analysis or the reaction was stopped by adding Krebs-Ringer bicarbonate buffer (Cat # K4002, Sigma-Aldrich) and cell pellets were resuspended in Krebs-Ringer bicarbonate buffer with 1% TCA (for ATP/ADP) or cold PBS (for NAD⁺/NADH). The ATP/ADP ratio was measured by Cellular ADP/ATP-Glo™ Assay (Promega AG), according to the instructions of the manufacturer. Lactate and the NAD⁺/NADH ratio were measured for luciferase activity by Lactate-Glo™ Assay or NAD/NADH-Glo™ Assay (Promega AG), according to the instructions of the manufacturer. Values were normalized to the total protein content.

**Degranulation assay**. Primary bone marrow-derived mouse neutrophils ($4 \times 10^6$ cells per ml) were resuspended in X-VIVO™ 15 medium, primed with GM-CSF and subsequently activated with C5a. For degranulation analysis of azurophil granule protein, in the final 5 min of GM-CSF priming, cytochalasin B (5 μM) was added to the cell suspension. Degranulation of azurophil granules was determined by measuring the increase in CD63 expression as a surrogate marker. Surface expression of CD63 was performed using PE-conjugated rat anti-mouse CD63 monoclonal antibody (clone NVG-2; Cat # 564222, BD Biosciences, dilution 1:50) and flow cytometry (FACSVerse, BD Biosciences)[59]. CD63 expression was quantified using FlowJo software (Tree Star, Ashland, OR, USA). For β-glucosaminidase activity, the reaction was stopped by adding ice-cold HBSS-0.1% BSA and the supernatants were collected following a centrifugation step (20 min, $425 \times g$, 4 °C). Cell pellets were lysed with 0.12% Triton X-100 for 10 min at room temperature. The substrate solution (5 mM p-nitrophenyl-2-acetamido-2-deoxy-α-D-glucopyranoside in 25 mM sodium citrate, pH 4.5) was added to the cell lysate and supernatant followed by incubation at 37 °C for 1 h. The reaction was stopped by adding 52 mM glycine-NaOH, pH 10.5, and the absorbance measured at 410 nm using a SpectraMax M2 plate reader (Bucher Biotech). The amount of β-glucosaminidase released in supernatants of neutrophils was expressed as a percentage of the total β-glucosaminidase[60].

**Transmission electron microscopy**. Neutrophils were centrifuged at $425 \times g$ and 4ºC for 20 min. The supernatants were removed and 1 ml cold 2.5% glutaraldehyde solution in 0.15 M HEPES was added to cell pellets. Electron microscopy images were taken at an acceleration voltage of 80 keV on a Philips CM100 electron microscope equipped with a FEI Morgagni digital camera and images analyzed using iTEM software. The numbers of mitochondria per cell and the morphometric analysis of mitochondria and cristae were carried out as previously described using the Imaris software[10,35,61].

**Cell death assay**. Cell death was assessed by uptake of ethidium bromide fluorescent dye (25 μM) and analyzed by flow cytometer (BD FACSCalibur)[2,21,62].

**Statistical analysis**. Analysis of all data was performed using the Prism software (GraphPad Software Inc., San Diego, CA, USA). All data are expressed as the means ± SEM. To compare groups, one-way ANOVA followed by Tukey's multiple comparisons test or the Kruskal–Wallis test was applied. Differences between many groups at two levels were tested by two-way ANOVA followed by Bonferroni post tests. $p$ values of <0.05 were considered statistically significant.

**Data availability**. All relevant data are available from the authors upon request.

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

## Acknowledgements

This work was supported by grants from the Swiss National Science Foundation to S.Y. (number 31003A_173215) and H.-U.S. (number 310030_166473). P.A. and D.S. are Ph.D. students of the Graduate School of Cellular and Biomedical Sciences of the University of Bern. Images were acquired on equipment supported by the Microscopy Imaging Centre of the University of Bern.

## Author contributions

D.S. and P.A. conceived, planned, and performed the study, analyzed and interpreted the data, and wrote the paper. A.F., C.B.J., and C.B. performed experiments; C.C. took clinical care of the ADOA patients; A.S. supported the genetic analysis of ADOA patients; L.G. supported the super-resolution multicolor imaging; M.E.S. and L.S. provided *Opa1*<sup>flox/flox</sup> mice and supported the mouse experiments; C.B., J.-M.N., and S.Y. provided experimental advice and edited the paper; H.-U.S. provided overall guidance, experimental advice and laboratory infrastructure and edited the paper; all authors read and approved the final manuscript.

## Additional information

**Competing interests:** The authors declare no competing interests.

