## [Peer Review File · Nature Communications]

Reviewers' comments:

Reviewer #1 (Remarks to the Author):

In this study, Amini and colleagues investigate the requirement of OPA-1 in neutrophil extracellular trap (NET) formation. The authors show that loss of OPA-1, specifically, results in NET production deficiency following stimulation with GM-CSF and C5a. As OPA-1 is the only one of the dynamin-GTPase family members that affects the formulation of NETs, the authors conclude that mitochondrial fusion-independent functions of OPA-1 are critical for NET formation. As OPA-1 KO neutrophils did not show significant mitochondrial translocation to the plasma membrane, which is a hallmark of NET formation¹, the authors asked whether microtubule or actin assembly was impaired in OPA-1 KO neutrophils. The authors observe a deficiency specifically in microtubule dynamics while actin polymerization and ROS production were unaffected following stimulation with GM-CSF and C5a. Additionally, microtubule assembly and NET formation in OPA-1 KO neutrophils is rescued by addition of ATP to the culture. Since OPA-1 has known roles in promoting the stability of mitochondrial respiratory complexes, the authors examined ATP production in WT and OPA-1 KO neutrophils. At baseline total ATP levels in neutrophils could be decreased by inhibition of complex I or glycolysis but not complex III or ATP synthase. This data led the authors to conclude that neutrophil ATP production is primarily glycolytic, as has been established in the literature, but that complex I plays a role in glycolytic ATP production. The authors then compared WT and OPA-1 KO neutrophils and found total ATP, NAD⁺, lactate levels and complex I activity at baseline were decreased in OPA-1 KO neutrophils. From this the authors hypothesize that OPA-1 stabilized complex I, which provides NAD⁺ critical for glycolytic ATP production. This ATP, in turn, is necessary for microtubule assembly that drives NET production in GM-CSF+C5a stimulated neutrophils. To test this hypothesis the authors pharmacologically inhibit complex I, ATP synthase or glycolysis function in WT neutrophils stimulated with GM-CSF and C5a and find that inhibition of complex I or glycolysis but not ATP synthase inhibits microtubule assembly and NET formation. Leading them to conclude that glycolysis and complex I function but not ATP synthase are necessary for neutrophil NET formation. The authors also observe that OPA-1 KO neutrophils are deficient in bacterial killing in vitro and in vivo, indicating the functional relevance of this pathway to neutrophils in the setting of infection. While the premise that OPA-1 stabilizes complex I which provides NAD⁺ for glycolytic production of ATP that is critical for microtubule assembly and NET formation is interesting, this reviewer believes that the data presented does not adequately support this hypothesis without significant revisions outlined below.

Major concerns:

1. Rotenone has already been shown to directly inhibit microtubule assembly^{2,3} independent of its effects on complex I. In order to confirm that NET formation requires complex I function as in figure 6A and B, and is not an off-target effect of the inhibitor I would like to see use of another inhibitor such as Piericidin A.
2. I have concerns about the evidence presented to support the claim that OPA-1 is critical for complex I production of NAD⁺ that is required for glycolytic ATP synthesis and subsequent NET formation. The authors observe decreased ATP levels (figure 5b), complex I function (figure 5c), NAD⁺ (figure 5d) and lactate levels (figure 5e) in OPA-1 KO neutrophils. However, all of these experiments were performed on neutrophils without GM-CSF+C5a stimulation. This supports the idea that ATP production may be deficient at baseline, but doesn't take into account that GM-CSF or C5a may rescue this deficiency. I would like to see if the differences are maintained when the cells are stimulated to make NETs by GM-CSF and C5a. In addition, the authors use decreased ATP and NAD⁺ levels per g protein in OPA-1 KO neutrophils to demonstrate OPA-1's requirement for complex I function. However, the ratio of NAD⁺/NADH and ATP/ADP or AMP ratios are more informative of the redox and energetic state of the cells than NAD⁺ or ATP alone. Additionally, use of the Seahorse XF analyzer to compare baseline glycolytic function in WT and OPA-1 KO neutrophils as well as assay how inhibition of complex I, complex III or complex V affects glycolysis in neutrophils would provide greater support for the idea that OPA-1 is critical for complex I function and subsequent glycolytic ATP production in neutrophils.
3. The exogenous addition of ATP rescued NET formation and microtubule assembly, however, this

does not confirm the authors claim that ATP produced by glycolysis in the neutrophils is sufficient to rescue NET production. First, I would like to see that addition of ATP rescues microtubule assembly and NET formation in WT neutrophils treated with inhibitors as in figures 6a and b. Beyond using ATP to rescue NET formation, I would like to see experiments to rescue NAD⁺ levels downstream of complex I NAD⁺ production. Nicotinamide mononucleotide (NMN) has been used in vitro and in vivo to rescue NAD⁺ levels. This could be used with OPA-1 KO to test whether increasing NAD⁺ levels rescues microtubule assembly (Figure 6a) and NET formation (Figure 6b) as well as in vitro (Figure 7a) and in vivo (Figure 7c-e) bacterial killing.

4. The authors claim that they have demonstrated that "complex I activity indirectly regulates glycolysis, which is required for microtubule network and functional NET formation." Indicating they have uncovered an underlying role for the mitochondria in NET formation, however, NETs can be induced by a variety of stimuli, of which the authors only test one. To conclude that OPA-1 is critical for NET formation more globally as implied by the authors assertion above, I would like to see evidence that OPA-1 KO neutrophils do not form NETs in response to other stimuli such as PMA or ribonucleoprotein immune complexes.

5. In figure 7f it is difficult to identify NETs in these images as the PI is oversaturated and covers the signal and I am unclear how the quantification works as it is not covered in the materials and methods. In addition a clearer way to stain to confirm that what you are seeing is NETs and not background staining is outlined in 4. Utilizing three color fluorescence (DAPI, myeloperoxidase (MPO) and histone H1) would be a preferred method to identify and quantify NET formation in vivo.

Minor Concerns:

1. Text and figure 1a legend states that c.1140G>A/p.E380E mutation leads to exon 13 skipping however the reference (Schimpf et al. 2006) states that this is exon 11 skipping. Please clarify.
2. In figure 7e MPO is used as a surrogate marker to support the claim that "there is no defect in the neutrophil recruitment to the lungs of OPA1N mice." Enumerating total and neutrophil cell numbers by flow cytometry on lung digests would be a better method to demonstrate this point.
3. Text line 347-348: "Recently, increasingly, researchers report release of mtDNA during NET formation, supporting our earlier findings." Sentence should be changed to be grammatically correct.

References

1. Lood, C. et al. Neutrophil extracellular traps enriched in oxidized mitochondrial DNA are interferogenic and contribute to lupus-like disease. *Nature Medicine* 22, 146–153 (2016).
2. Brinkley, B. R., Barham, S. S., Barranco, S. C. & Fuller, G. M. Rotenone inhibition of spindle microtubule assembly in mammalian cells. *Exp Cell Res* 85, 41–46 (1974).
3. Srivastava, P. & Panda, D. Rotenone inhibits mammalian cell proliferation by inhibiting microtubule assembly through tubulin binding. *FEBS Journal* 274, 4788–4801 (2007).
4. Röhm, M. et al. NADPH oxidase promotes neutrophil extracellular trap formation in pulmonary aspergillosis. *Infect. Immun.* 82, 1766–1777 (2014).

Reviewer #2 (Remarks to the Author):

This study describes a mitochondrial deficiency that appears to affect NET formation. While it further validates the role of mitochondria in NET formation, there is a complete disconnect between the in vitro and in vivo data that must be reconciled.

1)The authors never show that *P.aureginosa* induces mitochondrial NETs. In fact the whole study uses C5a and GM-CSF and then in mice they use bacteria. There is a complete disconnect between the two situations. One almost certainly does not reflect the other. It is odd that this type of disconnect pervades the NET literature. If the aim is to show that these knockout mice have an inability to clear *pseudomonas* infections than use the same stimulus in vitro namely *Pseudomonas* and not some set of molecules that may or may not have much to do with *pseudomonas*.

Is the Lyz2 promoter specific for neutrophils. Should it not knockout this molecule in macrophage?

Why is the son with the mutation not susceptible to infections. This does not make any sense.

In summary it is not clear why sometimes the authors used C5a and GM-CSF and then in mouse they use *P.aureginosa*. Why not the same stimulus throughout. There is no demonstration that NETs are involved in vivo.

Reviewer #3 (Remarks to the Author):

This is an interesting study based on the observation that by silencing the major proteins of the fission/fusion machinery, only the lack of OPA1 affects a specific function of neutrophils, namely the capability of formation of neutrophil extracellular traps (NETs), which exert an antibacterial action. This is a peculiar antibacterial strategy based on the release of mtDNA, which becomes instrumental to the formation of NETs.

In order to understand better the mechanism behind OPA1 involvement in the NETs formation, the authors first investigate 2 patients with a heterozygous haploinsufficiency mutation leading to DOA, a father and his son, obtaining the puzzling result that while the father had a normal DNA release from his blood-isolated neutrophils, this was deficient in the son.

I can immediately comment that this result is very weak and unclear. First, there is no known evidence that DOA patients have anything wrong with their antibacterial defenses. Second, the OPA1 expression is reduced but not abolished and it is a matter of fact that in non-syndromic DOA the only affected cells are the retinal ganglion cells and their optic nerve-forming axons. Third, an exploratory study if the neutrophils are affected in DOA patients cannot rely on only two subjects with opposite results. Either the authors run a proper screening in a adequately powered cohort of patients or otherwise this result remains meaningless and should be deleted from the paper. Finally, just out of curiosity, given the known variability in DOA penetrance and clinical severity, did the defective neutrophil function of the son match a more severe optic atrophy as compared with the father?

The study goes on establishing a mouse model of knockout OPA1 only in neutrophils (*Opa1 Δ* mice), which confirms the deficient phenotype of impaired NETs formation. Taking advantage of this model the authors demonstrate in a series of experiments that the deficient NETs formation depends on disruption of the microtubule network, which in turn seems the result of an impaired glycolytic ATP production due to reduced respiratory complex I activity. In fact, the authors rescue the phenotype by exogenous supplementation of ATP, whereas evoke the phenotype by rotenone-induced complex I inhibition as well as 2-DG induced inhibition of glycolysis. The authors also exclude a role played by apoptosis, ROS overproduction and, more surprisingly, apparently they fail to observe mtDNA depletion in the *Opa1 Δ* neutrophils.

This last point bothers me the most, given the unequivocal documentation in multiple models that lack of OPA1 induces profound mtDNA depletion (just consult the literature), which interestingly could be another reason for failing NETs formation. Without getting too deep into the technicality of the real time PCR-based test, why the authors chose 18s rRNA as nuclear reference gene, which is multicopy, instead of having as almost all protocols for mtDNA copy number assessment a single copy nuclear reference gene, so that the ratio is between 2 copies per cell nDNA reference against multiple mtDNA genomes? To confirm their unexpected result with mtDNA the authors could also test the abundance and distribution of mtDNA nucleoids in the *Opa1 Δ* neutrophils, as an

independent approach. A further experiment would be assessing TFAM abundance, as this is the major protein collapsing the mtDNA genomes into a nucleoid.

Another point that I would like to see investigated is the complex I deficiency. Is complex I assembled, and also super-assembled with complex III? A thorough investigation of why complex I is deficient in Opa1NΔ neutrophils is mandatory to better understand and link this to the proposed deficient glycolysis. Overall, is the disruption of the microtubule network truly due to an ATP deficiency or are there other possible OPA1-mediated mechanisms beyond having merely a bioenergetic failure?

Minor points:

Lines 71-73: I do not think that Mitofusins are anchored on the inner mitochondrial membrane with OPA1, please correct.

Lines 225-227: reference 12 as supportive for reduced ATP synthesis in skeletal muscle of DOA patients is inappropriate, please use something more appropriate.

In conclusion, the authors show that their mouse model is more prone to infection with *Pseudomonas aeruginosa*, as consequence of neutrophil malfunction despite their increased recruitment. This study has undoubtedly points of interest, but I think there is much work to be done to clarify the critical issues I have raised.

NCOMMS-17-24255-T: Point-by-point reply

Reviewer #1

1. Rotenone has already been shown to directly inhibit microtubule assembly (2,3) independent of its effects on complex I. In order to confirm that NET formation requires complex I function as in figure 6A and B, and is not an off-target effect of the inhibitor, I would like to see use of another inhibitor such as Piericidin A.

Answer: The previous publications mentioned (Brinkley et al., *Exp. Cell Res.* 1974 and Srivastava et al., *FEBS J.* 2007) reported that rotenone inhibited mitosis by preventing microtubule assembly. Both reports failed to demonstrate the molecular mechanism for the disruption of mitosis, but speculated that rotenone physically binds to tubulin. In this study, we provide data indicating that the reason for the lack of proper microtubule network formation in terminally differentiated mouse and human neutrophils is a lack of “energy”. By adding ATP or, as suggested by the Reviewer, nicotinamide mononucleotide (NMN), microtubule assembly and NET formation were restored in Opa1-knockout neutrophils within 30 min of incubation (revised Fig. 4a and 4b, new Fig. 6a, and 6b *lower panels*). We have also followed this Reviewer’s suggestion and performed functional assays, such as dsDNA release and microtubule network assembly, in the presence of Piericidin A (PA) in order to avoid misleading conclusions owing to possible off-target effects of rotenone. As with rotenone, PA blocked microtubule network assembly and also inhibited NET formation (revised Fig. 6a and 6b). All new data are described in the text (pages 12 and 15).

2. I have concerns about the evidence presented to support the claim that OPA1 is critical for complex I production of NAD⁺ that is required for glycolytic ATP synthesis and subsequent NET formation. The authors observe decreased ATP levels (Figure 5b), complex I function (Figure 5c), NAD⁺ (Figure 5d) and lactate levels (Figure 5e) in OPA1 KO neutrophils. However, all of these experiments were performed on neutrophils without GM-CSF+C5a stimulation. This supports the idea that ATP production may be deficient at baseline, but doesn’t take into account that GM-CSF or C5a may rescue this deficiency. I would like to see if the differences are maintained when the cells are stimulated to make NETs by GM-CSF and C5a.

Answer: In the revised manuscript, we now present the ATP/ADP ratios (Fig. 5b), complex I activities (Fig. 5c), NAD⁺/NADH ratios (Fig. 5d) and lactate levels / μg protein (Fig. 5e) for both unstimulated and stimulated neutrophils. The results indicate that stimulation of neutrophils does not rescue a lack of ATP; in fact, it further reduces ATP levels, perhaps owing to the energy requirement for microtubule network rearrangements as previously suggested

(Bershadsky & Gelfand, *PNAS* 1981). All new data are described in the text (pages 13 and 14).

In addition, the authors use decreased ATP and NAD⁺ levels per g protein in OPA1 KO neutrophils to demonstrate OPA1's requirement for complex I function. However, the ratio of NAD⁺/NADH and ATP/ADP or AMP ratios are more informative of the redox and energetic state of the cells than NAD⁺ or ATP alone.

Answer: As Reviewer 1 requested, we now provide data demonstrating NAD⁺/NADH and ATP/ADP ratios, respectively (revised Fig. 5b and 5d and text, pages 13 and 14).

Additionally, use of the Seahorse XF analyzer to compare baseline glycolytic function in WT and OPA1 KO neutrophils as well as to assay how inhibition of complex I, complex III or complex V affects glycolysis in neutrophils would provide greater support for the idea that OPA1 is critical for complex I function and subsequent glycolytic ATP production in neutrophils.

Answer: The extracellular acidification rate (ECAR) and oxygen consumption rate (OCR) were analyzed using the Seahorse XF analyzer (new Suppl. Fig. 9a, b). Consistent with the lactate values measured, we found a general decrease in ECAR (an indicator of aerobic glycolysis). Acute inhibition of complex V (oligomycin) led to a slight increase of ECAR, whereas acute inhibition of complex I (rotenone) showed no effect or a tendency to a slight decrease in ECAR. In agreement with the limited OXPHOS activity, OCR exhibited a general decrease, which was most pronounced when respiration was stimulated maximally using the uncoupler FCCP. Furthermore, we provide new data obtained by measurements of complex I and complex III activities. These results indicate that Opa1-knockout neutrophils exhibit a defect in complex I, but not complex III activity (revised Fig. 5c). The discrepancies between the clearly reduced complex I activity and the only mildly affected OCR might be explained by the fact that mitochondria of neutrophils can receive electrons from glycolysis via glycerol-3-phosphate through the glycerophosphate dehydrogenase complex into the Q cycle (see new ref. 41, van Raam et al., *PLoS One* 2008). The text was modified according to the new data (pages 14 and 15).

3. The exogenous addition of ATP rescued NET formation and microtubule assembly, however, this does not confirm the authors claim that ATP produced by glycolysis in the neutrophils is sufficient to rescue NET production. First, I would like to see if addition of ATP rescues microtubule assembly and NET formation in WT neutrophils treated with inhibitors as in Figures 6a and b.

Answer: We followed the Reviewer's suggestion and now provide data demonstrating that exogenous ATP rescues microtubule network assembly as well as NET formation in activated

wild-type mouse neutrophils in the presence of rotenone, 2-DG or PA (revised Fig. 6a and 6b, *lower panels*; modified text, page 15).

Beyond using ATP to rescue NET formation, I would like to see experiments to rescue NAD⁺ levels downstream of complex I NAD⁺ production. Nicotinamide mononucleotide (NMN) has been used in vitro and in vivo to rescue NAD⁺ levels. This could be used with OPA1 KO to test whether increasing NAD⁺ levels rescues microtubule assembly (Figure 6a) and NET formation (Figure 6b) as well as in vitro (Figure 7a) and in vivo (Figure 7c-e) bacterial killing.

Answer: We now provide data that exogenous NMN also rescued microtubule network assembly and NET formation in Opa1-knockout neutrophils (revised Fig. 4a and 6b; revised Fig. 6a and 6b, *lower panels*; new Suppl. Fig. 7a and 7b; modified text, pages 12 and 16), as well as bacterial killing capacity (killing assays with both *E. coli* and *P. aeruginosa*) (revised Fig. 7a). However, we decided not to test NMN in an experimental *in vivo* system, as Reviewer 1 suggested, since the data might be inconclusive owing to the possibility that NMN would also affect the metabolism of other cell types (as reported previously by Yoshino et al., *Cell Metabol* 2011). Please note that Opa1 Δ mice exhibit homozygous deletion of Opa1 in myeloid cells only.

4. The authors claim that they have demonstrated that “complex I activity indirectly regulates glycolysis, which is required for microtubule network and functional NET formation.” Indicating they have uncovered an underlying role for the mitochondria in NET formation, however, NETs can be induced by a variety of stimuli, of which the authors only test one. To conclude that OPA1 is critical for NET formation more globally as implied by the authors assertion above, I would like to see evidence that OPA1 KO neutrophils do not form NETs in response to other stimuli such as PMA or ribonucleoprotein.

Answer: In the original manuscript, we had already provided data in which mouse neutrophils had been subjected to other stimuli, lipopolysaccharide (LPS) or bacteria (*E. coli*) (Fig. 1e). We have now followed the Reviewer’s suggestion and provide additional data for a variety of stimuli that have been reported to induce NET formation such as PMA, bacteria (*E. coli* and *P. aeruginosa*), and immune complexes (reported by Lood et al., *Nat Med* 2016) (new Suppl. Fig. 2c, new Suppl. Fig. 4c, as well as new Suppl. 7a and 7b). With all stimuli, we observe the same results: Activated Opa1-knockout neutrophils fail to assemble microtubule networks and lack the ability to form NETs.

5. In Figure 7f, it is difficult to identify NETs in these images as the PI is oversaturated and covers the signal. I am unclear how the quantification works as it is not covered in the Materials and Methods. In addition a clearer way to stain to confirm that what you are seeing are NETs and not background staining is outlined in (4). Utilizing three-color fluorescence

(DAPI, myeloperoxidase (MPO) and histone H1) would be a preferred method to identify and quantify NET formation in vivo.

Answer: We followed the Reviewer's advice and stained mouse lung tissue samples according to the protocol described by Röhm et al., *Infection Immunity* 2014. We now can clearly demonstrate the co-localization of granule protein (MPO) with extracellular DNA release (Fig. 7f, including newly added images showing NETs at higher magnification).

Minor Concerns:

1. *Text and Figure 1a legend state that the c.1140G>A/p.E380E mutation leads to exon 13 skipping; however the reference (Schimpf et al. 2006) states that this is exon 11 skipping. Please clarify.*

Answer: This point is now corrected in the manuscript on page 6.

2. *In Figure 7e, MPO is used as a surrogate marker to support the claim that "there is no defect in the neutrophil recruitment to the lungs of OPA1N mice." Enumerating total and neutrophil cell numbers by flow cytometry on lung digests would be a better method to demonstrate this point.*

Answer: Please note that we had provided data for increased myeloperoxidase activity in lung tissue extracts as a sign for increased neutrophil infiltration. This is an accepted method (Ref. 41, Benarafa et al., *J Exp Med* 2007). To confirm these data, we now provide additional data, generated with a new set of stained tissue samples, which were analyzed using an automated slide scanner. Here, we quantified the percentage of MPO positive cells against the total number of propidium iodide (PI) positive cells, using the algorithm provided by the manufacturer (3DHISTECH slide scanner, Quant Center software, with the Cell Count module (new Fig. 7f, second right panel).

3. *Text line 347-348: "Recently, several researchers report the release of mtDNA during NET formation, thus supporting our earlier findings." Sentence should be changed to be grammatically correct.*

Answer: We thank the Reviewer and have corrected the text accordingly (page 18).

Reviewer #2

This study describes a mitochondrial deficiency that appears to affect NET formation. While it further validates the role of mitochondria in NET formation, there is a complete disconnect between the in vitro and in vivo data that must be reconciled.

1) *The authors never show that P. aeruginosa induces mitochondrial NETs. In fact the whole study uses C5a and GM-CSF and then in mice they use bacteria. There is a complete*

disconnect between the two situations. One almost certainly does not reflect the other. It is odd that this type of disconnect pervades the NET literature. If the aim is to show that these knockout mice have an inability to clear pseudomonas infections, then use the same stimulus in vitro, namely Pseudomonas, and not some set of molecules that may or may not have much to do with pseudomonas.

Answer: NETs could be induced by several physiological stimuli including priming with GM-CSF followed by C5a activation. Our original manuscript also reported findings with other NET-release stimuli such as *E. coli* and lipopolysaccharide (LPS) (Fig. 1e), and demonstrated *in vitro* bacterial killing by NETs using *E. coli* (Fig. 7a). In the revised manuscript, we now provide additional data depicting microtubule network and NET formation stimulated with PMA or *P. aeruginosa* infection, as well as with immune complexes (as previously reported by Lood et al., *Nat Med* 2016 – ref. 29) (new Suppl. Fig. 2c, new Suppl. Fig. 4c, as well as new Suppl. 7a and 7b). With all stimuli, we observe the same results: Activated Opa1-knockout neutrophils fail to assemble microtubule networks and lack the ability to form NETs.

2) Is the Lyz2 promoter specific for neutrophils. Should it not knockout this molecule in macrophages?

Answer: Lyz2 also affects the macrophage population. Therefore, to dissect the molecular mechanism of NET formation and its dependence on a neutrophil metabolism owing to Opa1 deficiency were performed in *in vitro* experimental setups using pure neutrophil populations. However, under *in vivo* conditions, a possible role of OPA1-defective macrophages cannot be excluded. We have indicated this fact in the Discussion (page 20).

Why is the son with the same mutation not susceptible to infections. This does not make any sense.

Answer: Unfortunately, the frequency of infections of the patients described in our work has not been recorded. However, after additional discussion with the mother of the affected son, we learned that her son suffered from recurrent pneumonia infections. Since we have no documentation about these infections, we decided not to include this information in the manuscript. However, we now mention our findings so that clinicians should be aware of the possibility that ADOA patients may exhibit an increased risk to infections (page 20).

In summary, it is not clear why sometimes the authors used C5a and GM-CSF and then in mouse they use P. aeruginosa. Why not the same stimulus throughout. There is no demonstration that NETs are involved in vivo.

Answer: As mentioned above, we now present data using multiple stimuli, including *P. aeruginosa*, *in vitro*. We also demonstrate NET involvement under *in vivo* conditions using an experimental *P. aeruginosa* infection model.

Reviewer #3

I can immediately comment that this result is very weak and unclear. First, there is no known evidence that DOA patients have anything wrong with their antibacterial defenses.

Answer: OPA1 is a mitochondrial protein. Mitochondria have been shown to play an important role in innate immunity (Arnoult et al., *EMBO reports* 2011). In addition, there are several recent publications directly connecting the role of OPA1 and the mitochondrial modulation of immune function(s). For example, it has been reported that OPA1 is essential for the development of memory T cells after infection (Buck et al., *Cell* 2016). Furthermore, deletion of Opa1 in muscle cells caused systemic inflammation (Tezze et al., *Cell Metabolism* 2017). Thus, published evidence indicates that OPA1 impacts the immune system. Unfortunately, the frequency of infections in ADOA patients has not yet been investigated. However, the previously published work as well as our current data clearly point to the possibility of an unmet need when caring for ADOA patients. Therefore, we now raise this possibility so that clinicians are aware that ADOA patients may exhibit an increased risk of infections (page 20).

Second, OPA1 expression is reduced but not abolished, and it is a matter of fact that in non-syndromic DOA, the only affected cells are the retinal ganglion cells and their optic nerve-forming axons.

Answer: Non-syndromic ADOA is defined in textbooks as affecting only the optic nerve-forming axons. However, it is interesting to consider that such patients also exhibit a heterozygous deficient OPA1 gene in all cell types. Why some OPA1 mutations are manifest as syndromic, and others as non-syndromic, is still not clear.

Third, an exploratory study to see if neutrophils are affected in DOA patients cannot rely on only two subjects with opposite results. Either the authors run a proper screening in an adequately powered cohort of patients or otherwise this result remains meaningless and should be deleted from the paper.

Answer: ADOA is a rare, heterogeneous disease, making it difficult to collect a large cohort of ADOA patients within a reasonable time period. Our findings should just demonstrate that ADOA patients *can* exhibit defects in neutrophil functions. Furthermore, since ADOA patients still exhibit a wild-type allele with a normal OPA1 gene, we have chosen to generate conditional OPA1 knockout mice, in which we deleted the Opa1 gene homozygously in the mature myeloid cell population (Opa1 Δ mice).

Finally, just out of curiosity, given the known variability in DOA penetrance and clinical severity, did the defective neutrophil function of the son correlate with a more severe optic atrophy as compared with the father?

Answer: The son, 7 years at the time of blood analysis, had already displayed clinical signs of optic atrophy. The father had exhibited optic atrophy at much older age.

The study goes to establish a mouse model of knockout OPA1 only in neutrophils (Opa1 Δ mice), which confirms the deficient phenotype of impaired NET formation. Taking advantage of this model the authors demonstrate in a series of experiments that the deficient NET formation depends on disruption of the microtubule network, which in turn seems the result of an impaired glycolytic ATP production due to reduced respiratory complex I activity. In fact, the authors rescue the phenotype by exogenous supplementation of ATP, whereby they evoke the same phenotype by rotenone-induced complex I inhibition as well as 2-DG induced inhibition of glycolysis. The authors also exclude a role played by apoptosis, ROS overproduction and, more surprisingly, apparently they fail to observe mtDNA depletion in the Opa1 Δ neutrophils. This last point bothers me the most, given the unequivocal documentation in multiple models that lack of OPA1 induces profound mtDNA depletion (just consult the literature) which interestingly could be another reason for failing NET formation.

Answer: Previously published work indeed suggests that OPA1-deficiency results in a disorder of mtDNA maintenance (Chen et al., *Cell* 2010; Amati-Bonneau et al., *Brain* 2008). However, in blood leukocytes, OPA1-deficient patients had significantly higher mtDNA copy numbers compared to controls (Sitarz et al., *Neurology* 2012). In fact, it was suggested that increased mtDNA copy number in leukocytes of ADOA patients might act as a compensatory bioenergetic mechanism (Sitarz et al., *Neurology* 2012). In the original version of our report, we confirmed these findings and observed no difference in the content of mtDNA between wild-type and Opa1 knockout mouse neutrophils. In the new version of our manuscript, we additionally analyzed mtDNA copy numbers in the ADOA patients and in control individuals and found no differences (new Fig. 1d). The reason for the normal content of mtDNA in neutrophils could be the relatively short half-life of these cells. In addition, in our knock-out mouse model, depletion of *Opa1* in mouse neutrophils occurs only within the last three days of terminal neutrophil differentiation (Cre_Lyz2-dependent). Previous reports have already demonstrated that four days after Cre-mediated *Opa1* ablation, mtDNA copy numbers were unaffected even in non-neutrophilic cells (Cogliati et al., *Cell* 2013; Zhang et al., *Mol Biol Cell* 2011). Hence, a detectable natural depletion of mtDNA within this short period of time cannot be expected. Nevertheless, we have further quantified the number of mitochondria and amount of mtDNA, as well as the expression of mitochondrial transcription factor A (TFAM), which indirectly correlates with mtDNA levels, and found no difference between control and Opa1 knockout mouse neutrophils (page 11; revised Fig. 3c and new Fig. 3d).

Without getting too deep into the technicality of the real time PCR-based test, why do the authors chose 18s rRNA as nuclear reference gene, which is multicopy, instead of having, as almost all protocols for mtDNA copy number, assessment of a single copy nuclear reference gene, so that the ratio is between 2 copies per cell nDNA reference against multiple mtDNA genomes?

Answer: We thank Reviewer 3 for this justified criticism and now provide new data for both mouse and human neutrophils. We analyzed patients' neutrophils for mtDNA content (ATP8 and D-loop) compared to single copy nuclear DNA ($\beta 2$ microglobulin, B2M) (new Fig. 1d). We have also repeated the mouse genomic qPCR for mouse single copy nuclear DNA (mB2M) and mouse mtDNA (*mt-Co1*) (revised Fig. 3c). We found no significant differences between OPA1-deficient and control neutrophils, in both mouse and human systems. We, therefore, conclude that the lack of mtDNA release in Opa1-deficient neutrophils is not due to a reduced complement of mtDNA.

To confirm their unexpected result with mtDNA, the authors could also test the abundance and distribution of mtDNA nucleoids in the Opa1 Δ neutrophils, as an independent approach. A further experiment would be assessing TFAM abundance, as this is the major protein collapsing the mtDNA genomes into a nucleoid.

Answer: We followed the Reviewer's recommendation and stained mouse neutrophils from wild-type and Opa1 Δ mice for TFAM protein expression (new Fig. 3d). We did not observe any differences in TFAM expression between control and Opa1 knockout neutrophils, indirectly indicating that they have equivalent amounts of mtDNA.

Another point that I would like to see investigated is the complex I deficiency. Is complex I assembled, and also super-assembled with complex III? A thorough investigation of why complex I is deficient in Opa1 Δ neutrophils is mandatory to better understand and link this to the proposed deficient glycolysis. Overall, is the disruption of the microtubule network truly due to an ATP deficiency or are there other possible OPA1-mediated mechanisms beyond having merely a bioenergetic failure?

Answer: We now provide new data obtained by measurements of complex I (CI) and complex III (CIII) activities (new Fig. 5c, new Suppl. Fig. 9) (pages 14-15). The results of these experiments clearly show that Opa1 knockout neutrophils exhibit a defect in CI, but not CIII activity. To confirm our conclusion that OPA1 knockout results in a bioenergetics failure, we added in the new version of our manuscript, besides exogenous ATP, also exogenous NMN (precursor of NAD⁺) (revised Fig. 4a and 4b, revised Fig. 6a and 6b, *lower panels*, as well as new Suppl. Fig. 7a and 7b). Both ATP and NMN restored the capacity to form the microtubule network and NET formation, pointing to a bioenergetics failure. In view of the findings above

and the already published observation that human neutrophils do not have a supercomplex organization, we decided not to analyze supercomplex assembly (see new ref. 41, van Raam et al., *PLoS One* 2008).

Minor points:

Lines 71-73: I do not think that Mitofusins are anchored on the inner mitochondrial membrane with OPA1, please correct.

Answer: We thank Reviewer 3 for raising this point. We have corrected the sentence as follows: “OPA1 is anchored to the mitochondrial inner membrane, and together with MFN1 and MFN2, located in the outer mitochondrial membrane, controlling the mitochondrial fusion process” (page 4).

Lines 225-227: reference 12 as supportive for reduced ATP synthesis in skeletal muscle of DOA patients is inappropriate, please use something more appropriate.

Answer: We have now corrected the sentence and added a new reference as follows: “.....muscle cells from adult ADOA patients displayed reduced mitochondrial ATP synthesis, suggesting dysregulation of the mitochondrial metabolism as a consequence of OPA1 dysfunction” (page 11). We have replaced the reference with something more appropriate (Ref. 36; Lodi et al., *Arch Neurol* 2011).

REVIEWERS' COMMENTS:

Reviewer #1 (Remarks to the Author):

paper is acceptable

Reviewer #2 (Remarks to the Author):

Done